# A connexin/*ifi30* pathway bridges HSCs with their niche to dampen oxidative stress

Pietro Cacialli ⬤ [1], Christopher B. Mahony ⬤ [1,3], Tim Petzold[1], Patrizia Bordignon[2], Anne-Laure Rougemont ⬤ [2] & Julien Y. Bertrand ⬤ [1✉]

Reactive oxygen species (ROS) represent a by-product of metabolism and their excess is toxic for hematopoietic stem and progenitor cells (HSPCs). During embryogenesis, a small number of HSPCs are produced from the hemogenic endothelium, before they colonize a transient organ where they expand, for example the fetal liver in mammals. In this study, we use zebrafish to understand the molecular mechanisms that are important in the caudal hematopoietic tissue (equivalent to the mammalian fetal liver) to promote HSPC expansion. High levels of ROS are deleterious for HSPCs in this niche, however this is rescued by addition of antioxidants. We show that *Cx41.8* is important to lower ROS levels in HSPCs. We also demonstrate a new role for *ifi30*, known to be involved in the immune response. In the hematopoietic niche, *Ifi30* can recycle oxidized glutathione to allow HSPCs to dampen their levels of ROS, a role that could be conserved in human fetal liver.

[1] University of Geneva, Faculty of Medicine, Department of Pathology and Immunology, Geneva 4, Switzerland. [2] Division of clinical pathology, Geneva University Hospitals, Geneva 4, Switzerland. [3] Present address: Institute of Cancer and Genomic Sciences, University of Birmingham, Edgbaston, Birmingham, UK. ✉email: Julien.bertrand@unige.ch

In all vertebrates, hematopoiesis occurs in two main waves, starting with the emergence of primitive hematopoietic cells, and culminating with the specification of hematopoietic stem and progenitor cells (HSPCs) from the aortic hemogenic endothelium[1,2]. The very few HSPCs that are produced then colonize a niche where they can expand, the fetal liver and the caudal hematopoietic tissue (CHT)[3], in mammals and zebrafish, respectively. This embryonic niche is the only niche where HSPCs actively expand during the whole life of an animal[4], therefore, understanding the microenvironmental signals required at the non-cell-autonomous level for HSPC expansion may allow better expansion of human HSPCs ex vivo, improving our current clinical protocols. Many studies have pointed out the role of several cytokines in this process, which led to the use of KIT-L, FLT3L, and ThPO to promote human HSPC expansion[5–9]. However, the mechanisms controlling detoxification of HSPCs from metabolic wastes have been understudied. Reactive oxygen species (ROS) are a by-product of HSPC metabolism and their excess is toxic, inducing their senescence[10,11]. Although low levels of ROS appear to be required for proper HSPC activity, including their specification from the hemogenic endothelium[12] and hematopoietic reconstitution after transplantation[13], an excess of ROS can lead to impaired HSPC function[10]. Antioxidant molecules are therefore important to maintain cellular ROS levels in cells.

Reduced glutathione (GSH) is considered to be one of the most important scavengers of ROS. GSH plays an important role in protecting cells from oxidative damage and in maintaining redox homeostasis. GSH is a tripeptide composed of cysteine, glutamic acid, and glycine, and is synthetized by γ-glutamylcysteine synthetase (γ-GCS) and glutathione synthetase at the cellular level[14]. The γ-GCS-catalyzed production of γ-glutamylcysteine is the first and rate-limiting step in de novo GSH synthesis, as the availability of cysteine is limited[15]. This step is inhibited by GSH through a negative feedback loop, a mechanism that is central to the regulation of cellular GSH concentrations. Ultimately, two molecules of GSH can be oxidized into GSSG to eliminate one molecule of $H_2O_2$ and produce water. However, the cellular reservoir of GSH is limited and oxidized GSSG must be recycled into GSH, which can be achieved by the action of several enzymes[16,17].

Gap junctions are formed by the assembly of connexins from two adjacent cells, which allows them to exchange small metabolites. In the mouse bone marrow, previous studies have shown that HSPCs could transfer ROS directly to their hematopoietic microenvironment through Cx43[18]. In this context, stress induced by repeated injections of 5FU led to HSPCs senescence in Cx43-deficient animals, as they were accumulating ROS. The authors of this study also showed that Cx43 was similarly necessary in the microenvironment[18]. Whether or not a similar process involving gap junctions occurs in the fetal liver remains to be determined. However, this raises the hypothesis of a non cell-autonomous mechanism for ROS detoxification.

Previous in vitro data have shown the possible involvement of gilt/ifi30 in this process[19,20]. Gilt/ifi30 was the first identified and characterized gene encoding a lysosomal redox enzyme which catalyses thiol–disulfide exchange reactions[21] and was initially described as IP30[22]. Gilt/ifi30 encodes a 35-kDa precursor glycoprotein which is transported via the mannose 6-phosphate receptor to the endocytic pathway where the N- and C-terminal propeptides are cleaved into the 30-kDa mature form[23,24]. Both precursor and mature Gilt reduce disulfide bonds optimally at an acidic pH[24]. The cDNA encoding gilt has been identified in a variety of species including human, mouse, zebrafish[25], large yellow croaker[26], amphioxus[27], and abalone[28]. In mammals, IFI30 plays an important role in major histocompatibility complex class II-restricted antigen processing by reducing the disulfide bonds of phagocytosed proteins in the endocytic compartment[29]. In mice, the expression of Gilt/Ifi30 was also reported in mature T cells and some fibroblasts[19]. Zebrafish ifi30 encodes a protein very similar to the mammalian Gilt/Ifi30, with a thioredoxin-like C-X-X-C motif that forms the reductase active site[30] and mutations of either or both cysteines in the active-site abolish thiol-reductase activity of Gilt/Ifi30 [24].

In the present study, we demonstrate a new physiological role for ifi30— expressed by the hematopoietic microenvironment— independent of its role during the immune response. Here we show that ifi30, expressed by the vascular niche, plays an important role in recycling GSH in order to neutralize ROS produced by HSPCs during their expansion in the zebrafish CHT. This cooperation between HSPCs and their niche is possible through connexin channels, in particular Cx41.8. Indeed, their deficiency results in the accumulation of ROS in HSPCs, while the deficiency in ifi30 results in the accumulation of ROS in the microenvironment. We also show that the expression of IFI30 is conserved in human embryos, as hematopoietic progenitors in the human fetal liver are always closely associated with IFI30+ cells, which we identify to be macrophages.

## Results

**Oxidative stress induces a defect of HSPC proliferation**. To evaluate the impact of ROS during HSPC expansion in the CHT, we treated kdrl:mCherry;cmyb:GFP embryos with 3 mM hydrogen peroxide ($H_2O_2$) for three hours starting at ~45 hpf, and scored the number of $GFP^+$ cells (HSPCs) embedded within their $mCherry^+$ (endothelial) niche. This concentration was chosen after we tested the toxicity of $H_2O_2$ on zebrafish embryos (Supplementary Fig. 1a). We observed a loss of HSPCs in the CHT as early as 48 hpf (Fig. 1a, b and Supplementary Fig. 1b). This was accompanied by a decrease in HSPC proliferation at 48 hpf, as measured by anti-phospho-Histone 3 (pH3) staining after exposure to $H_2O_2$ (Fig. 1c, d), showing that ROS have a toxic effect on HSPCs in the CHT. This phenotype was completely rescued by co-treating embryos with reduced GSH (Fig. 1e, f), at 10 μM, a non-toxic concentration that rescues oxidative stress as measured by Cell-Rox fluorescence (Supplementary Fig. 1a, c). Therefore, the accumulation of ROS is deleterious for HSPCs, as expected, but not for endothelial cells, as their number was unchanged (Supplementary Fig. 1d). This defect can be rescued by the addition of antioxidant molecules, such as GSH. In parallel, we also tested the impact of other ROS enhancer compounds on HSPCs in the CHT at 48 hpf. We found that after 10 h of menadione treatment (10 μM), the number of HSPCs was reduced at 48 hpf (Supplementary Fig. 2a, b). We also evaluated the impact of the glutathione synthesis inhibitor buthionine sulfoximine (BSO, 10 μM) and oxidized glutathione GSSG (10 μM) treatments on HSPCs in the CHT. Both treatments resulted in a significant decrease in the number of cmyb-expressing cells in the CHT (Supplementary Fig. 2c–f), which was most significant upon treatment with GSSG. These results confirm that oxidative stress is deleterious for HSPCs during their expansion in the CHT, and prompted us to further investigate the role of the glutathione pathway in this process.

**ifi30 is expressed in the niche and controls HSPC expansion**. Previous studies have shown that Ifi30 was primarily expressed in antigen-presenting cells but also in fibroblasts in the adult mouse[19]. In the zebrafish embryo, we found ifi30 expression in CHT endothelial cells (ECs), more specifically in ECs of the posterior cardinal vein, as determined by in situ hybridization at 24, 33, and 48 hpf (Fig. 2a, b). This was confirmed by qPCR

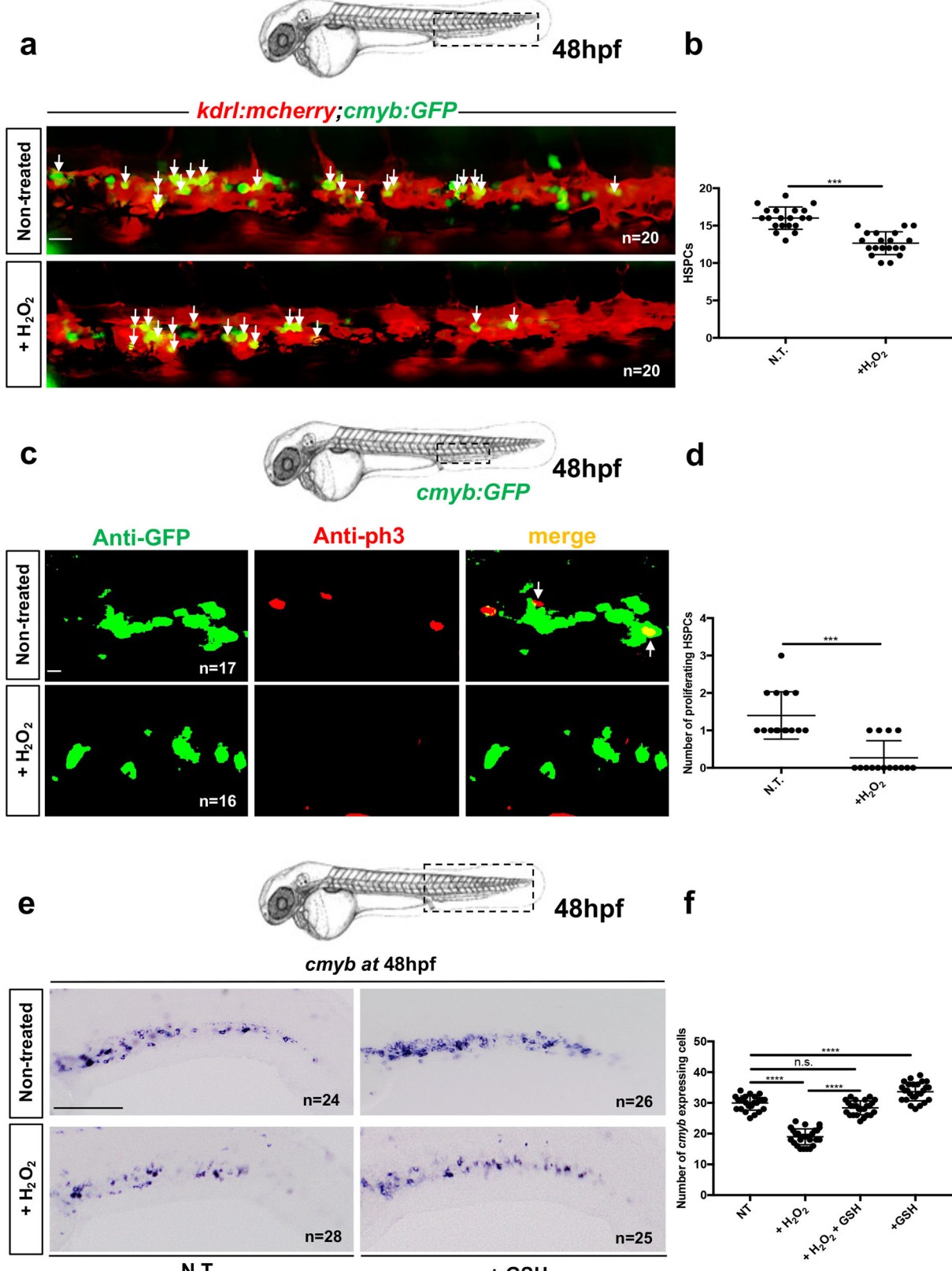

**Fig. 1 Oxidative stress induces a defect in HSPC proliferation in the CHT niche at 48 hpf. a** Confocal imaging in the CHT of 48 hpf *kdrl:mCherry/cmyb:GFP* embryos. Embryos non-treated (NT) or incubated with $H_2O_2$ (3 mM). **b** Quantification of HSPCs associated with ECs. Statistical analysis: unpaired two-tailed *t* test, ***P < 0.001. Center values denote the mean, and error values denote s.e.m. **c** Anti-GFP and pH3 immunostaining of NT or $H_2O_2$-treated (3 mM) *cmyb:GFP* embryos. **d** Quantification of proliferating HSPCs; statistical analysis: unpaired two-tailed *t* test, ***P < 0.001. Center values denote the mean, and error values denote s.e.m. **e** *cmyb* expression at 48 hpf in NT embryos, and embryos treated with $H_2O_2$ (3 mM) or GSH (10 μm). **f** Quantification of *cmyb*-expressing cells in the CHT at 48 hpf. Statistical analysis was performed using a one-way ANOVA with Tukey–Kramer post hoc tests, adjusted for multiple comparison, ****P < 0.0001; (n.s.) non-significant P = 0.15. Center values denote the mean, and error values denote s.e.m. Scale bars: 50 μm (**a**); 25 μm (**c**); 100 μm (**e**).

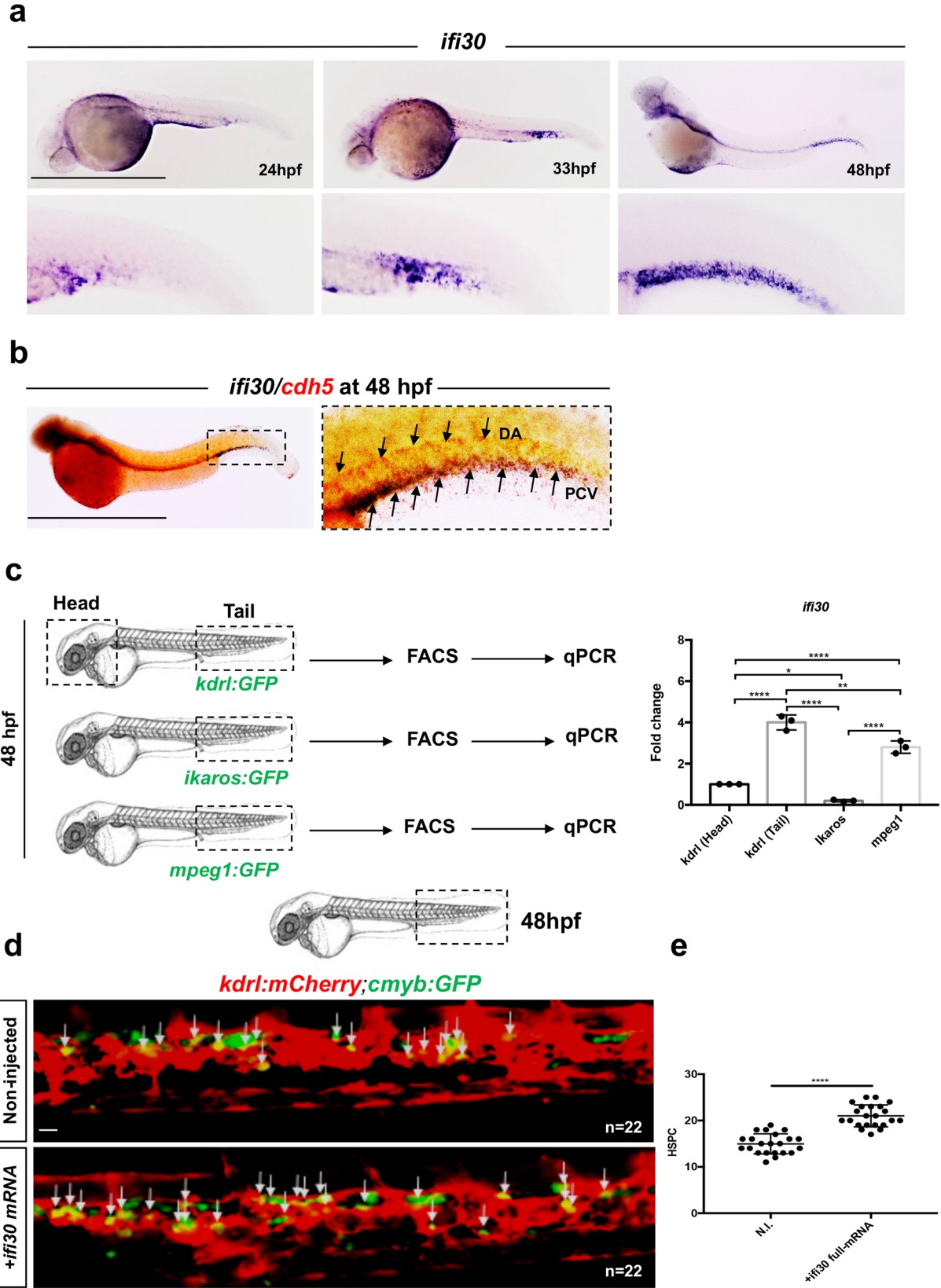

performed on purified *kdrl:GFP⁺, ikaros:GFP⁺, mpeg1:GFP⁺* cells (Supplementary Fig. 3a–e), showing high enrichment of *ifi30* in caudal ECs, as well as in macrophages, although to a lesser extent (Fig. 2c). The overexpression of *ifi30* mRNA increased the number of HSPCs in *kdrl:mCherry;cmyb:GFP* at 48 hpf (Fig. 2d). This increase was maintained at 4dpf and correlated with an

increase of *rag1* expression at 4.5dpf in injected embryos (Supplementary Fig. 4a–c). However, this gain-of-function did not increase *runx1* expression at 28hpf, therefore excluding any contribution of *ifi30* to HSPCs emergence (Supplementary Fig. 4d). Similarly, we observed no change in the number of HSPCs at 36hpf when *ifi30* full-length mRNA was injected in

**Fig. 2 ifi30 is expressed in CHT endothelial cells. a** WISH against *ifi30* at 24, 33, and 48 hpf. **b** Double WISH at 48 hpf against *ifi30* (purple) and *cdh5* (red), black arrows indicate the expression of *cdh5* in dorsal aorta (DA) and the overlapping expression with *ifi30* in the posterior cardinal vein (PCV). **c** qPCR data examining *ifi30* expression (fold change relative to expression in *kdrl:GFP*+ head subset) in FACS-sorted cells from *kdrl:GFP*, *ikaros:GFP*, and *mpeg1:GFP* embryos. GFP+ cells from dissected heads and tails were sorted from 48 hpf embryos (for each transgenic line have been pooled $n = 20$ embryos), and each experiment was repeated independently three times. Statistical analysis was performed using a one-way ANOVA with Tukey–Kramer post hoc tests, adjusted for multiple comparisons, *$P = 0.013$; **$P = 0.001$; ****$P < 0.0001$. Center values denote the mean, and error values denote s.e.m. **d** Imaged area in the tail at 48 hpf, as indicated by the black dotted line. Confocal imaging in the CHT of *kdrl:mCherry/cmyb:GFP* embryos in non-injected and *ifi30*-full mRNA injected embryos. The white arrows indicate *kdrl* and *cmyb* double-positive cells. **e** Quantification of HSPCs associated to ECs. Statistical analysis: unpaired two-tailed *t* test ****$P < 0.0001$. Center values denote the mean, and error values denote s.e.m. Scale bars: 200 μm (**a**, **b**); 50 μm (**d**).

*kdrl:mCherry;cmyb:GFP* embryos (Supplementary Fig. 4e, f). These results suggest that *ifi30* is primarily expressed in ECs, and can increase the proliferation of HSPCs in the CHT niche only and does not affect HSPC specification. To confirm this hypothesis, we generated *kdrl:Gal4;UAS:ifi30* embryos to specifically overexpress *ifi30* in ECs (Fig. 3a). First, we confirmed that our system worked efficiently by examining the expression of *ifi30* in double-transgenic embryos. *Kdrl:Gal4* single-positive embryos showed normal expression of *ifi30* (in the CHT), whereas *kdrl:Gal4;UAS:ifi30* embryos showed ectopic expression of *ifi30* in all the vasculature (Fig. 3b). Next, we confirmed that the hemogenic endothelium was not impaired in these embryos, as *runx1* and *cmyb* were normal in *kdrl:Gal4;UAS:ifi30* double-positive embryos at 28 and 36hpf, respectively, indicating no change in HSPC emergence when *ifi30* is overexpressed in ECs (Supplementary Fig. 5a, b). However, *kdrl:Gal4;UAS:ifi30* embryos showed an expansion of HSPCs in the CHT as determined by *cmyb* expression at 4dpf and 48 hpf (Fig. 3c–e) as well as an increase in the T-lymphocyte marker *rag1* at 4.5dpf (Supplementary Fig. 5c, d), without affecting the number of ECs in the CHT (Supplementary Fig. 5e, f). Altogether, these data show that *ifi30*, expressed by the vascular microenvironment, can control HSPC expansion in the CHT. Previous data have suggested an antioxidant role for *ifi30* in the mouse, so we then tested whether *ifi30* could rescue the loss of HSPCs induced by oxidative stress.

**Ifi30 controls the redox status of the CHT niche.** Before testing our hypothesis, we evaluated the impact of oxidative stress on *ifi30* expression by treating embryos with different ROS enhancers, such as $H_2O_2$ and menadione. Next, we also investigated the impact of the glutathione synthesis inhibitor buthionine sulfoximine (BSO) and oxidized glutathione GSSG. After treatment, we scored *ifi30* expression in treated embryos by qPCR at 48 hpf. We found a significant increase in *ifi30* expression following all pro-oxidant treatments tested (Supplementary Fig. 6a, b). *Ifi30* levels were particularly enhanced by the addition of GSSG, which was also confirmed by in situ hybridization (Supplementary Fig. 6c). Next, to test the potential antioxidant activity of *ifi30*, we used two approaches. First, we injected wild-type embryos with *ifi30*-mRNA at different concentrations before challenging them with $H_2O_2$ (3 mM) for three hours starting at ~45 hpf. We observed that the *cmyb* signal in the CHT was rescued proportionally to the concentration of *ifi30* mRNA injected (Supplementary Fig. 6d, e). Next, we challenged *kdrl:Gal4;UAS:ifi30* zebrafish embryos with $H_2O_2$ 3 mM for three hours starting at ~45 hpf (Fig. 3e, f). As expected, *cmyb* expression decreased in *kdrl:Gal4* single-positive embryos at 48 hpf in the CHT, but the HSPC number was maintained in *kdrl:Gal4;UAS:ifi30* embryos (Fig. 3f). These results confirm our hypothesis that the overexpression of *ifi30* in the vascular niche can rescue the loss of HSPCs after ROS induction in the CHT. We next examined the consequences of *ifi30*-deficiency on HSPCs, and used the uncharacterized *ifi30^{sa19758}* mutant line which presents a point mutation in exon 2, turning

the first cysteine of the catalytic site into a premature stop (Supplementary Fig. 7a, b). By in situ hybridization, we found that *ifi30^{−/−}* embryos exhibit a defect in definitive hematopoiesis as early as 48 hpf (Fig. 4a, b), which was maintained at 4dpf (Supplementary Fig. 8a). However, no change in *runx1* and *cmyb* expression were detected at 28 and 36 hpf, respectively, confirming that *ifi30* does not play a role at the level of the hemogenic endothelium (Supplementary Fig. 8b, c). Morpholino-mediated knockdown of *ifi30* expression completely phenocopied *ifi30^{−/−}* mutant embryos (Fig. 4a, b; Supplementary Fig. 7c, d; Supplementary Fig. 8a–c). Primitive erythropoiesis, vasculogenesis, and myelopoiesis were completely normal in *ifi30^{−/−}* mutants (Supplementary Fig. 9a–c). However, *ifi30^{−/−}* embryos exhibited a decrease in *mfap4* expression (macrophage marker) at 48 hpf (Supplementary Fig. 9d, e). To confirm our observations, we used time-lapse confocal imaging to follow the behavior of HSPCs in the CHT of *kdrl:mCherry;cmyb:GFP ifi30*-morphants. While the number of HSPCs augmented in control embryos between 42 and 48 hpf (Supplementary Movie 1; Supplementary Fig. 9f), their number remained unchanged in *ifi30*-deficient embryos (Supplementary Movie 2; Supplementary Fig. 9f). These results suggest that the absence of *ifi30* in the niche generates a HSPC proliferation defect in the CHT. To confirm this hypothesis, we injected control-morpholino and *ifi30*-morpholino in *cmyb:GFP* embryos and stained for both GFP and pH3 to quantify proliferating HSPCs. At 48 hpf, *ifi30*-morphants showed a significant decrease in the number of proliferating HSPCs (*cmyb:GFP*+/pH3+) within the CHT (Supplementary Fig. 10a, b). This phenotype was rescued by injecting *ifi30* mRNA (Supplementary Fig. 10c, d). Given the potential role of *ifi30* in maintaining redox homeostasis, we measured the levels of ROS in the CHT of control and *ifi30*-morphants at 48 hpf, by using the CellROX fluorescent probe. We could not detect any signal in control embryos, however, *ifi30*-morphants showed high levels of ROS in CHT-ECs (Fig. 4c), which was also confirmed by cytometry (Fig. 4d–g; for FACS strategy Supplementary Fig. 3f–l). This phenotype was also rescued by *ifi30* mRNA injection (Supplementary Fig. 10e, f). Here again, the resulting loss of *cmyb* and/or *runx1* in *ifi30*-morphants at 48 hpf could be rescued by adding anti-oxidants such as GSH or N-Acetyl-Cysteine (NAC) (Fig. 4h, i; Supplementary Fig. 11a–c). Surprisingly, ROS did not accumulate in HSPCs (Supplementary Fig. 11d) but rather in ECs (Fig. 4d–g) as well as in macrophages (Supplementary Fig. 12a, b). Of note, the accumulation of ROS in macrophages caused their cell death, as detected by acridine orange staining (Supplementary Fig. 12c, d), in line with our observation that *ifi30* mutants showed less *mfap4* signal in the CHT (Supplementary Fig. 9d, e). In summary, *ifi30* is expressed by the niche to control the redox status of the CHT and to therefore allow HSPCs to proliferate.

**Connexin-deficiency increases ROS causing apoptosis in HSPCs.** It was previously suggested that mouse adult HSPCs can transfer ROS to their microenvironment through gap junctions[31]. To

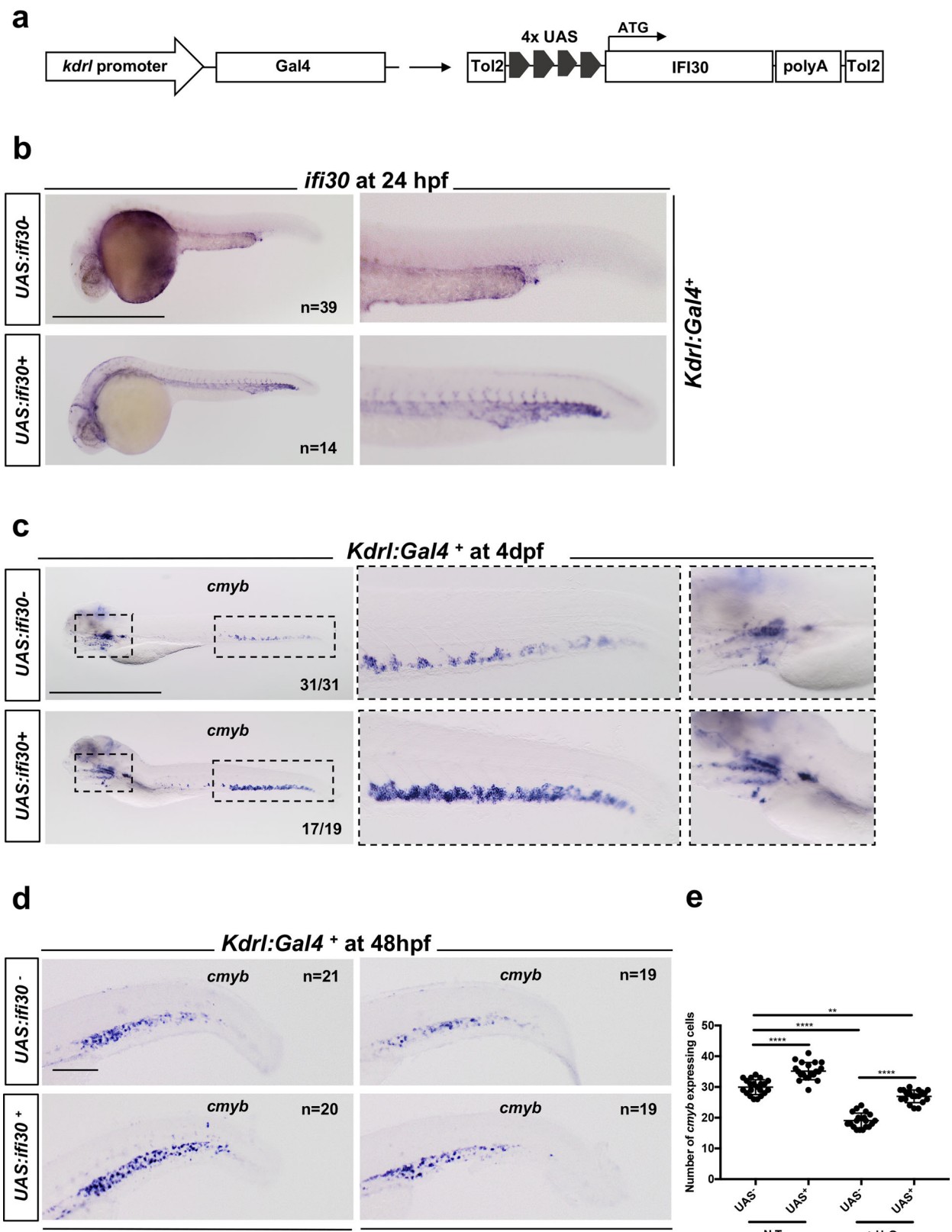

**Fig. 3 Ifi30 overexpression increases the number of HSPCs in the CHT and rescues the loss of HSPCs after ROS challenge. a** Experimental outline to generate double-transgenic *kdrl:Gal4;UAS:ifi30*. **b** WISH for *ifi30* at 24 hpf in *kdrl:Gal4+* embryos that were either *UAS:ifi30* negative (upper) or positive (lower). **c** WISH for *cmyb* at 4 dpf in *kdrl:Gal4+* embryos that were either *UAS:ifi30* negative (upper) or positive (lower). **d** WISH against *cmyb* at 48 hpf in *kdrl:Gal4+* embryos that were either *UAS:ifi30-* (upper) or *UAS:ifi30+* (lower), NT and $H_2O_2$-treated. **e** Quantification of *cmyb*-expressing cells. Statistical analysis: one-way ANOVA with Tukey–Kramer post hoc tests, adjusted for multiple comparisons, **$P = 0.013$; ****$P < 0.0001$. Center values of all statistical analyses denote the mean, and error values denote s.e.m. Scale bars: 200 μm (**b**, **c**), 100 μm (**d**).

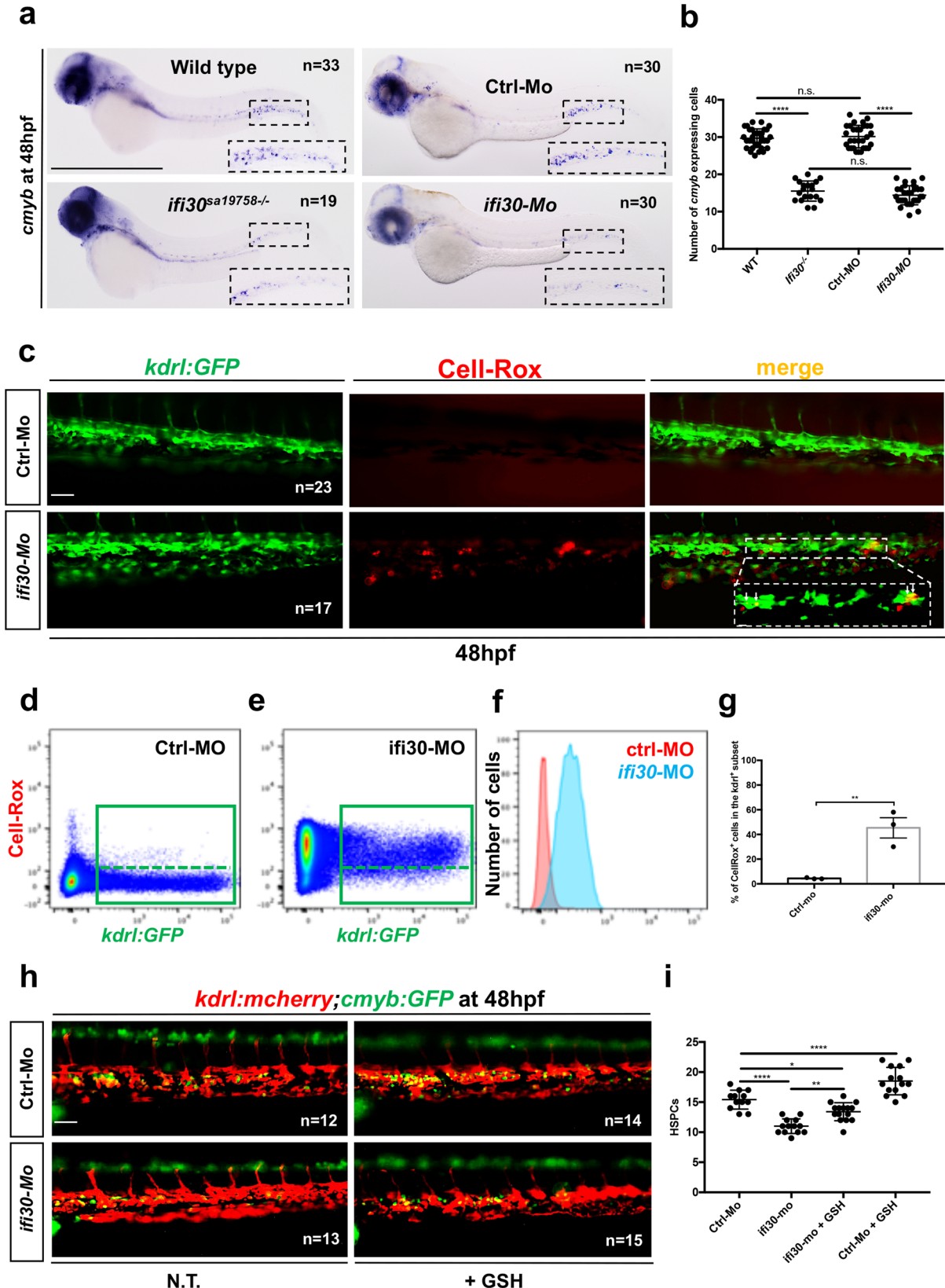

test a similar mechanism, we used two different approaches. First, we examined the status of HSPCs in *connexin41.8* (*cx41.8*) mutant embryos. This connexin has been previously described in zebrafish[32,33] and we recently reported that *cx41.8* plays an important role on ECs proliferation[34]. In zebrafish embryos, *cx41.8* is enriched in caudal ECs, as well as in HSPCs and macrophages, to a lesser extent, as scored by qPCR on sorted *kdrl:GFP*⁺, *ikaros:GFP*⁺, and *mpeg1:GFP*⁺ cells (Fig. 5a, b). *Cx41.8*⁻/⁻ mutants have not previously been characterized in terms of hematopoietic markers. The *cx41.8*⁻/⁻ mutant presents a point stop mutation (C-T) in exon 2, creating a premature STOP codon (Supplementary Fig. 13a). We found that *cx41.8*⁻/⁻ mutants showed normal HSPC specification

**Fig. 4 ifi30-deficiency increases ROS levels in ECs and induces a defect in HSPC proliferation. a** *cmyb* expression at 48 hpf in wild type and *ifi30*[−/−] embryos, after the injection of control or *ifi30*-morpholino. **b** Quantification of *cmyb*-expressing cells. Statistical analysis: one-way ANOVA, with Tukey–Kramer post hoc tests, adjusted for multiple comparisons, ****$P < 0.0001$, (n.s.) non-significant $P = 0.84$; $P = 0.32$. Center values of all statistical analysis denote the mean, and error values denote s.e.m. **c** Confocal imaging in the CHT of *kdrl:GFP*[+] cells (green), CellROX fluorescent probe (red) and merge (yellow). **d** Representative FACS plots showing dissociated zebrafish embryos after injection of control or (**e**) *ifi30*-morpholinos. FACS plots are gated on live cells. **f** Histogram plot showing the overlap of both samples, gated on *kdrl*[+] cells. **g** Quantification of the percentage of *kdrl:GFP*[+] cells affected by oxidative stress (ROS[+]) in *ifi30*-morphants (*ifi30*-MO) and controls (ctrl-MO) at 48 hpf. Data points are the mean of $n = 4$ biological replicates ± SD. Statistical analysis: unpaired two-tailed *t* test, **$P = 0.007$. Center values of all statistical analysis denote the mean, and error values denote s.e.m. **h** Fluorescence imaging in the CHT of *kdrl:mCherry;cmyb:GFP* embryos injected with control- and *ifi30*-MOs and treated with GSH. **i** Quantification of HSPCs associated with ECs. Statistical analysis: one-way ANOVA, with Tukey–Kramer post hoc tests, adjusted for multiple comparisons, *$P = 0.028$; **$P = 0.008$; ****$P < 0.0001$. The center values of all statistical analyses denote the mean, and error values denote s.e.m. Scale bars: 200 µm (**a**), 100 µm (**c**, **d**).

and emergence, as scored by *runx1* and *cmyb* staining at 28 and 36 hpf, respectively (Supplementary Fig. 13b, c). However, we found that *cx41.8*[−/−] mutant embryos present a significant decrease in HSPCs in the CHT at 60 hpf and 72 hpf, as scored by *cmyb* expression (Supplementary Fig. 13d–f). The role of gap junctions has previously been described in the mouse bone marrow, where Cx43-deficient HSPCs showed high levels of exhaustion after 5FU-induced exposure[18]. Furthermore, the authors showed that connexins regulate stress-induced hematopoietic regeneration following the transfer of 5FU-induced ROS from HSPC to the bone marrow niche and prevent ROS-p38-p16/INK4a-mediated HSPC cell-death[35]. To verify if a connexin deficiency would increase the level of ROS, we measured ROS in *cx41.8*[−/−] embryos compared with wild-type embryos at 60 hpf. We found a significant increase of CellROX-positive cells in the CHT of *cx41.8*[−/−] mutants (Supplementary Fig. 14a, b). We found that the majority of CellROX-positive cells also co-expressed *GFP* in *cx41.8*[−/−];*cmyb:GFP* embryos, confirming the hypothesis that the loss of connexin increases ROS levels in HSPCs (Supplementary Fig. 14c). Next, we tried to rescue the loss of HSPCs observed in the CHT of *cx41.8*[−/−] mutants by GSH addition. *Cx41.8*[−/−] embryos were supplemented with GSH (10 µM) between 48 and 60 hpf, which led to a rescue in HSPCs number, as scored by *cmyb* staining (Fig. 5c, d). These results highlight an important role for *cx41.8* during HSPC expansion in the zebrafish CHT.

Next, to evaluate the role of connexins in definitive hematopoiesis, we blocked these channels by treating *cmyb:GFP* embryos with heptanol or carbenoxelone[36,37]. As connexins are important for cell-cell communication in many tissues, we first tested the impact of this general inhibition on the cardiovascular system. We treated *myl7:DsRed* (myocardium) and *globin:GFP* (erythrocytes) transgenic embryos, with low concentrations (50 µM) of heptanol, to test the impact on blood circulation. This low concentration did not affect cardiac contractions or blood circulation (Supplementary Movies 3, 4, 5 and 6), whereas a higher concentration (1 mM) did (Supplementary Movies 7 and 8), as previously reported[37]. We therefore used heptanol or carbenoxelone at 50 µM. Following both treatments, HSPCs were depleted from the CHT, as measured by in situ hybridization (Supplementary Fig. 15a, b) and in *cd41:GFP* embryos at 48 hpf (Supplementary Fig. 15c). This was confirmed by in vivo time-lapse imaging, where *kdrl:mCherry; cmyb:GFP* embryos were imaged between 41 and 48 hpf (Supplementary Fig. 15d and Supplementary Movies 9 and 10). Whereas the number of HSPCs increased in non-treated embryos (Supplementary Movie 9; Supplementary Fig. 15d), we observed a loss of HSPCs in heptanol-treated embryos (Supplementary Movie 10; Supplementary Fig. 15d). Interestingly, we find that heptanol treatment phenocopied *cx41.8* mutants. To ascertain that Cx41.8 was the major connexin involved in this process, we treated wild type and *cx41.8*[−/−] embryos with heptanol (between 50 and 60 hpf), and we performed in situ hybridization for *cmyb* at 60 hpf. We scored the number of HSPCs in the CHT. We found a significant decrease in *cx41.8*[−/−] embryos, which was not

amplified by the additional heptanol treatment (Supplementary Fig. 16a–c), suggesting that Cx41.8 is the major connexin involved in this mechanism.

Next, we measured the levels of ROS in the CHT of heptanol-treated embryos. To assess the nature of the ROS accumulating cells, we treated *mpeg1:GFP* and *kdrl:GFP* embryos, to visualize CellROX in macrophages and ECs, respectively. However, unlike in *ifi30*-deficient embryos, ROS did never accumulate in macrophages or ECs (Supplementary Fig. 17a, b). But when we treated *cmyb:GFP* embryos with heptanol, we could see that CellROX staining was overlapping with GFP, indicating that ROS accumulated directly in HSPCs at 48 hpf (Fig. 5e), as expected from our previous experiment in *cx41.8*[−/−];*cmyb:GFP* embryos (Supplementary Fig. 14c). By using MitoSox-RED, we confirmed this increase of ROS production in HSPCs' mitochondria after exposure to heptanol (Supplementary Fig. 18a, b).

These high levels of ROS in HSPCs resulted in increased apoptosis in heptanol-treated *cmyb:GFP* embryos, as measured by TUNEL (Fig. 6a, b). Finally, the loss of HSPCs observed in the CHT of heptanol-treated *cmyb:GFP* embryos was also rescued by supplementation with GSH (Fig. 6c, d). However, this phenotype was not rescued by overexpressing *ifi30* in ECs (*kdrl:Gal4;UAS: ifi30*). This was expected since the accumulation of ROS at the cell-autonomous level could not be rescued by the anti-oxidant activity of *ifi30*, expressed in the niche (Supplementary Fig. 18c, d). To confirm the antioxidant activity of *ifi30*, we generated double-transgenic *gata2b:KalTA4;UAS:ifi30* embryos, to over-express *ifi30* directly in HSPCs. We found that after treatment with heptanol, the number of HSPCs in double-positive *gata2b: KalTA4;UAS:ifi30* embryos was restored to normal levels (Supplementary Fig. 18e, f). Taken together, we propose a mechanism (Fig. 6e), in which *ifi30*, expressed by CHT-ECs and, to a lesser extent by macrophages, plays a non-cell-autonomous role during HSPC expansion in the CHT niche.

**IFI30 expression is conserved in the human fetal liver.** *Ifi30* was initially described for its importance during antigen presentation on MHCII molecules. In order to validate this new role in the hematopoietic niche, we investigated the expression of *IFI30* and *CD117* in the human fetal liver, by antibody staining. We observed that the number of *IFI30*[+] cells mirrored the number of hematopoietic progenitors (*CD117*[+]), and that both subsets peaked between 12 and 14 gestational weeks (i.e. between 14 and 16 weeks of amenorrhea) (Fig. 7a, b), during which the hematopoietic activity of the human fetal liver also peaks[38]. Moreover, tight associations between human HSPCs and *IFI30*[+] cells were present in hematopoietic foci of the fetal liver (Fig. 7c). In human fetal liver, the expression of *IFI30* was restricted to macrophages, as shown by *IFI30/CD68* double staining performed at 13 gestational weeks (Fig. 7d). These data suggest that the expression of *IFI30* has been conserved from the zebrafish CHT to the human fetal liver.

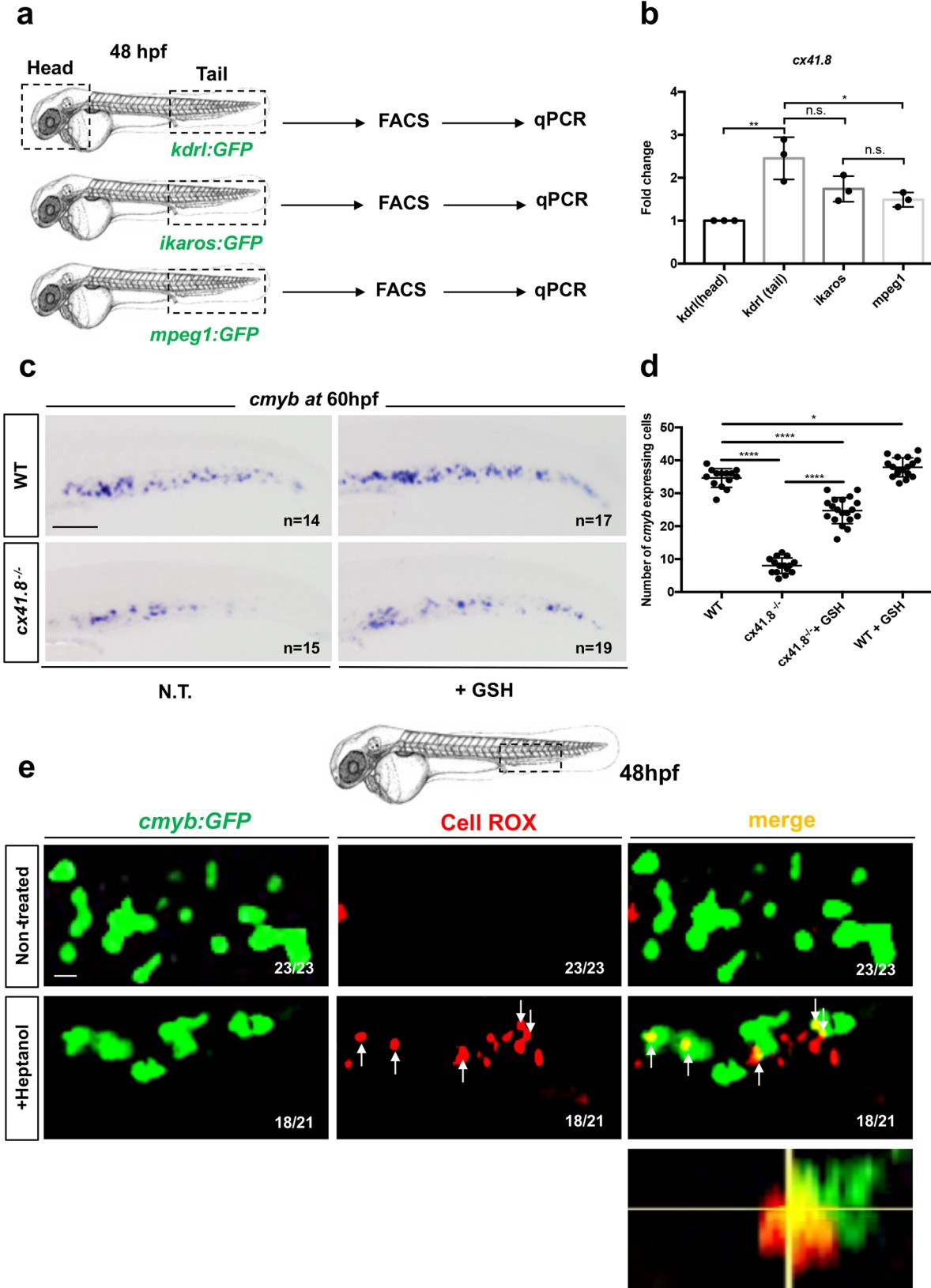

## Discussion

Several studies have reported that HSPCs are sensitive to oxidative stress as they reside in a hypoxic niche and are maintained in a quiescent state. Indeed, an increase of ROS levels can induce defects promoting HSPC senescence, ultimately causing premature HSPC aging as well as cell death. Our data confirm that an increase of ROS generates a defect in HSPC proliferation in the CHT niche of zebrafish embryos, which can be rescued by the addition of antioxidant molecules. GSH acts as an important antioxidant, but its cellular quantity is usually limited, which means that oxidized glutathione (GSSG) must be continuously recycled in order for cells to efficiently eliminate ROS. Many

**Fig. 5 Cx41.8−/− mutant embryos show a decrease in HSPCs which is rescued by antioxidant treatment. a** qPCR data examining *cx41.8* expression (fold change relative to expression in *kdrl:GFP*+ head subset) in FACS-sorted cells from *kdrl:GFP, ikaros:GFP* and *mpeg1:GFP* embryos. GFP+ cells from dissected heads and tails were sorted from 48 hpf embryos. **b** For each transgenic line have been pooled n = 20 animals, and each experiment was repeated independently three times. Statistical analysis was completed using a one-way ANOVA, with Tukey–Kramer post hoc tests, adjusted for multiple comparisons *P = 0.018; **P = 0.001; (n.s.) non-significant P = 0.26; P = 0.74. **c** *cmyb* expression at 60 hpf in wild type and *cx41.8−/−* embryos, and after GSH treatment. **d** Quantification of *cmyb*-expressing cells. Statistical analysis: one-way ANOVA, with Tukey–Kramer post hoc tests, adjusted for multiple comparisons, *P = 0.045; ****P < .0001. Center values denote the mean, and error values denote s.e.m. **e** Imaged area in the tail at 48 hpf, as indicated by the black dotted line. Confocal imaging of the CHT at 48 hpf of *cmyb:GFP*+ cells (green), CellROX fluorescent probe (red) and merge, in NT and after heptanol treatment *cmyb:GFP* embryos. Orthogonal projection CellROX+ *cmyb:GFP*+ double-positive cell after heptanol treatment. Scale bars: 100 μm (**c**); 25 μm (**e**).

enzymes have been proposed to act as GSSG reducers, including *IFI30/GILT*[20], which is mainly known for its role in antigen presentation[29]. In fact, it has been demonstrated that *Ifi30* knock-out mice cannot efficiently establish an immune response against pathogens as antigen-presenting cells cannot stimulate T lymphocytes[29,39]. Moreover, it was shown that these mice harbor high levels of cellular ROS and oxidized GSSG, as well as mitochondrial defects[20]. This has potentially been further explained by in vitro experiments showing involvement of *Ifi30*, in the balance of GSH/GSSG in mouse fibroblasts[19].

In our in vivo study on zebrafish embryos, we show that *ifi30* plays an important, newly identified role in the CHT niche. First, we show that *ifi30* is specifically expressed in ECs of the CHT niche. EC overexpression of *ifi30* results in an increase in the number of HSPCs in the CHT, and rescues the loss of HSPCs following challenge with ROS treatment. The novel finding of this study is therefore that *ifi30* functions independently of its immunological role and plays a role that was previously underestimated, as a regulator of the redox status in the hematopoietic niche. We show that *ifi30*, expressed in the niche, acts as an antioxidant, and behaves as such, since its expression is highly increased after an oxidative stimulus. As a thiol-reductase, *ifi30* can reduce protein disulfide bonds contained in oxidized glutathione GSSG produced by HSPCs into reduced GSH. However, since *ifi30* is expressed only in CHT-ECs and macrophages, this raises the question of the accessibility of HSPC-derived metabolites (ROS or oxidized GSSG). Transportation of these metabolites through cellular channels implies a high proximity between HSPCs and niche vascular ECs, something which has been previously shown by electron microscopy and termed "HSPC cuddling"[3]. Moreover, it was recently suggested that in the mouse bone marrow, adult HSPCs transfer ROS to their microenvironment through gap junctions[31,35]. In the present study, we have identified a connexin that is important for the function of the zebrafish hematopoietic niche. In particular, we have shown that *cx41.8−/−* mutants present a significant decrease in HSPCs in the CHT niche between 48 and 60 hpf, as well as embryos treated with heptanol or carbenoxelone. In these connexin-inhibited embryos, we show that HSPCs accumulate ROS at the cell-autonomous level, inducing their apoptosis, emphasizing the role of gap junctions in this process. To understand the interaction between HSPCs and their niche, several lines of evidence must be examined. The loss of HSPCs (due to connexin- or *ifi30*-deficiencies) can be rescued by the addition of NAC or GSH, which suggests that the glutathione pathway represents the major ROS scavenging pathway in the zebrafish CHT. In this pathway, a superoxide dismutase (*sod*) converts the superanion $O_2.^{-}$ into $H_2O_2$, which is then converted into $H_2O$ by a GSH peroxidase (*gpx*), that uses 2 molecules of GSH to form GSSH and 2 molecules of $H_2O$. When gap junctions are blocked, HSPCs accumulate ROS and enter apoptosis, but they can be rescued when they overexpress *ifi30*, which is only expressed in the microenvironment. This suggests that HSPCs can perform the first steps of the detoxification (using *sod* and a *gpx* enzyme that needs to be identified), but cannot recycle oxidized glutathione (GSSG),

a task that is mainly performed by the vascular niche. Indeed, as the amount of reduced GSH is limited in each cell, a failure to recycle GSSG would result in the accumulation of ROS, which we observe in Cx-deficient embryos. This raises the hypothesis of an exchange of reduced and oxidized glutathione exists between HSPCs and their microenvironment, through gap junctions.

Finally, we have demonstrated that this newly identified role of *ifi30* in the zebrafish CHT is also conserved in human fetal liver. In particular, we observe that *IFI30* is expressed in macrophages that are in close contact with hematopoietic progenitors (*CD117*+) in the fetal liver. These data reinforce the importance of macrophages to assist HSPCs during their embryonic life. While adult bone marrow macrophages have been described to retain HSPCs in the adult mouse bone marrow[40–43], the importance of macrophages for HSPC migration from the aorta to the CHT has been recently highlighted in the zebrafish embryo[44,45]. While sterile inflammation has proven to be essential to HSPC emergence and expansion[40–42] our data show a further example of how common factors of the immune toolbox have been reassigned to the establishment and maturation of the hematopoietic system.

## Methods

**Ethical statement.** Fish were raised according to FELASA and swiss guidelines[46]. No authorization was required since all experiments were performed before 5 days post fertilization. All efforts were made to comply to the 3R guidelines. The human sample study protocol was approved by the Commission Cantonale d'Ethique de la Recherche (CCER:2020-00582), University of Geneva, Switzerland. Informed written consent for both the autopsy and the further use of coded material for research purposes was obtained from the parents. Fetuses were recruited according to the following criteria: parental written approval to the use of samples for research purposes, gestational age, absence of liver pathology upon histological evaluation, adequate morphology, and in particular no tissue autolysis.

**Zebrafish husbandry.** AB* zebrafish strains, along with transgenic strains and mutant strains, were kept in a 14/10 h light/dark cycle at 28 °C. Embryos were obtained as described previously[47]. In this study we used the following transgenic lines: *Tg(kdrl:Gal4)*[bw9][48], *Tg(cmyb:GFP)*[zf16][49], *Tg(kdrl:Has.HRASmCherry)*[s896][50], *Tg(kdrl:GFP)*[s843][51], *Tg(mpeg1:mCherry)*[gl23][52], *Tg(mpeg1:GFP)*[gl22][52], *Tg(fli1a:nls-mCherry)*[ubs1053], *Tg(gata2b:KalTA4)*[sd32][54], *Tg(globin:GFP)*[cz332][55], *Tg(myl7:DsRED)*[s879][50], and the mutant line *leo*[t1/t1] *(referred to as Cx41.8−/−)* [33].

**Generation of transgenic animals.** For *Tg(UAS:ifi30)* fish generation, a Tol2 vector containing 4xUAS promoter, the sequence for *ifi30* (including STOP codon), and a poly-adenylation signal sequence was generated by sub-cloning. Zebrafish embryos were injected with 25 pg of the final Tol2 vector, and with 25 pg Tol2 transposase mRNA. Injected F0 adults were mated to AB zebrafish, and F1 offspring were screened to assess germline integration of the Tol2 construct. Starting with the F2, adults were crossed to the *Tg(kdrl:Gal4)* and/or *Tg(gata2b:KalTA4)* to perform experiments. After WISH, embryos were genotyped by PCR. Genotyping primers are listed in Supplemantaryemental Table S1.

**Identification of ifi30 mutant line.** The *ifi30*[sa19758] mutant line possesses a point stop mutation C-T in exon 2, in the first cysteine of the catalytic site. The *ifi30*[sa19758] line is available at the ZFIN repository (ZFIN ID: ZDB-GENE-030131-8447) from the Zebrafish International Resource Center (ZIRC). The *ifi30*[sa19758] used in this study was subsequently outcrossed with WT AB* for clearing of potential background mutations derived from the random ENU mutagenesis from which this line was derived. The *ifi30*[sa19758] homozygous mutant is referred in the paper as *ifi30−/−* and was obtained by incrossing our *ifi30*[sa19758]/AB strain. Genotyping was performed by PCR of the *ifi30* gene followed by sequencing. Genotyping primers are indicated in Supplemantaryemental Table S1.

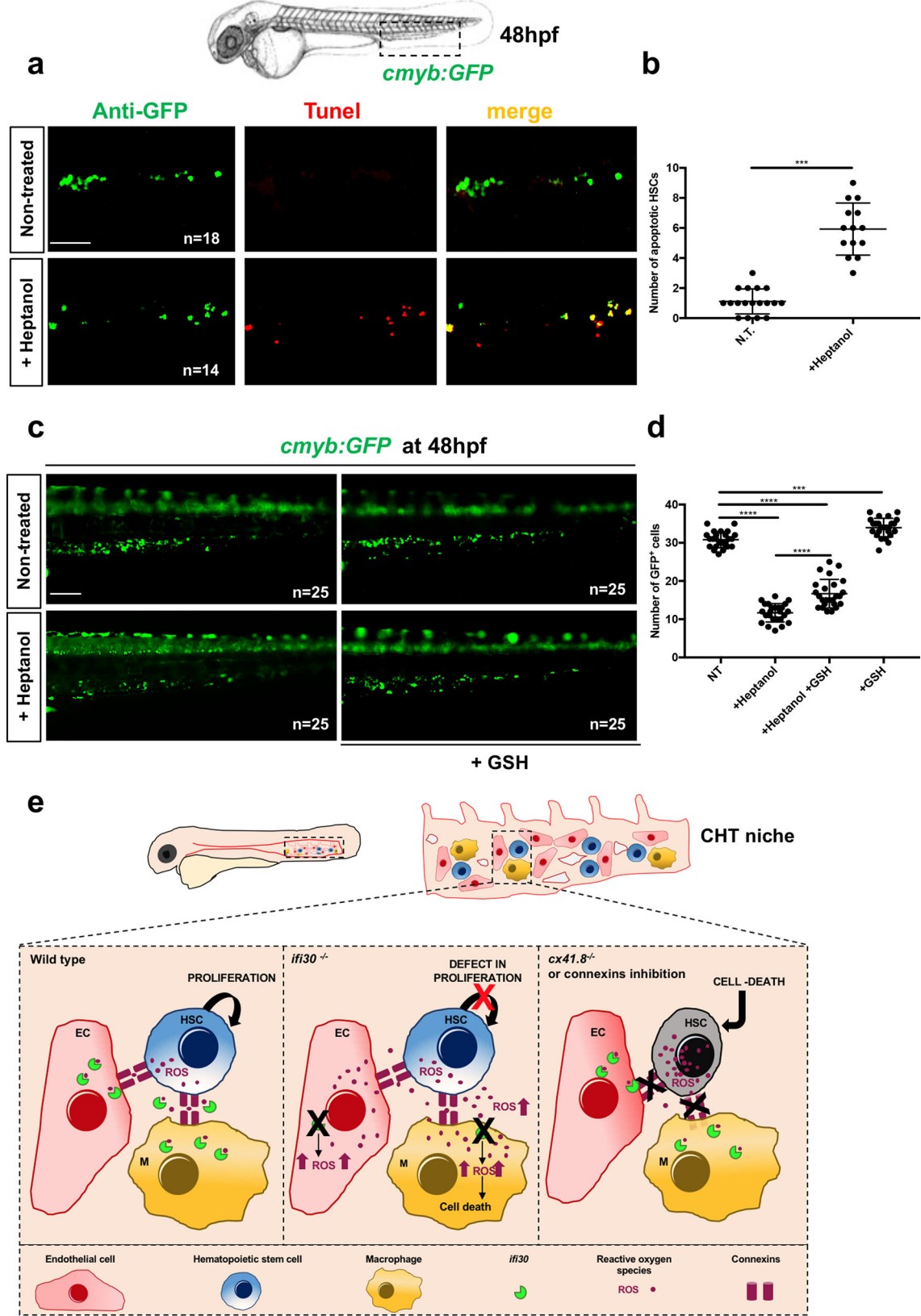

**Drug exposure and analysis**. The concentrations of all chemical treatments were chosen based on previous studies which established that these doses produced sublethal effects in zebrafish embryos, with no gross malformations[11,36,56]. All compounds used in these experiments were purchased from Sigma-Aldrich. Zebrafish embryos were exposed for three hours to 3 mM of $H_2O_2$ starting at ~45hpf. For chemical treatments, the embryos were exposed to 10 µm of NAC or L-glutathione reduced (GSH), Menadione, BSO, oxidized glutathione (GSSG) from 36hpf to 48 hpf in embryo E3 medium. New embryo medium with the fresh compound was administered daily until 4dpf. For heptanol and carbenoxolone treatments, wildtype zebrafish embryos were exposed to 50 µM doses from 36 to 48 hpf, and cx41.8 mutants from 48 to 60 hpf. After exposure, fish were fixed and the expression of different markers was tested by WISH. Embryos were phenotyped based on the expression of the markers and eventually genotyped (when treating ifi30 mutants or kdrl:gal4;UAS:ifi30 embryos).

**Fig. 6 Loss of HSPCs after connexin inhibition with drug treatment can be rescued by GSH treatment. a** Anti-GFP and TUNEL staining of NT and heptanol-treated *cmyb:GFP* embryos. **b** Quantification of the number of apoptotic HSPCs in both conditions. Statistical analysis was performed using an unpaired two-tailed *t* test ****P* < .001. Center values denote the mean and error values denote s.e.m. **c** Fluorescence imaging of NT and heptanol-treated (50 μM) *cmyb:GFP* embryos and after GSH treatment. **d** Quantification of *cmyb:GFP*+ cells. Statistical analysis: one-way ANOVA with Tukey–Kramer post hoc tests, adjusted for multiple comparisons, ****P* = 0007; *****P* < .0001. Center values denote the mean and error values denote s.e.m. **e** HSPCs directly communicate with ECs and macrophages (MΦ) in the embryonic niche through connexin channels. Under normal conditions *ifi30* reduces oxidized glutathione GSSG into reduced GSH, maintaining low levels of ROS in the CHT/HSPCs. The deficiency of *ifi30* increases GSSG levels and increases ROS in the CHT niche, generating a defect of HSPC proliferation. The connexin-deficient embryos (cx41.8 mutants or treated with connexin inhibitors) present ROS accumulation directly in HSPCs, inducing their cell death. Scale bars: 100 μm (**a–c**).

**Synthesis of full-length *ifi30* mRNA**. mRNA was reverse transcribed using mMessage mMachine kit SP6 (Ambion) from a linearized pCS2+ vector containing PCR-amplified product. After transcription, RNA was purified by phenol-chloroform extraction. Oligonucleotide primers used are listed in Supplemantaryemental Table 1.

**Morpholino injections**. The *ifi30*-morpholino oligonucleotide (MOs) and control-MO were purchased from GeneTools (Philomath, OR) and are listed in Supplemantaryemental Table 2. MO efficiency was tested by reverse transcription-polymerase chain reaction (RT-PCR) from total RNA extracted from ~10 embryos at 48 hpf using primers listed in Supplemantaryemental Table S2. In all experiments, 12 ng of *ifi30*-MO were injected per embryo.

**Whole-mount in situ hybridization and analysis**. Digoxigenin and fluorescein-labeled probes *cmyb, runx1, cdh5, pu.1, gata1, flk1, mfap4,* and *rag1* were previously described[57]. Whole-mount in situ hybridization (WISH) was performed on 4% paraformaldehyde-fixed embryos following the protocol described in ref. [58]. All injections were repeated three separate times. Analysis was performed using the unpaired Student *t*-test or ANOVA Multiple comparison-test (GraphPad Prism). Embryos were imaged in 100% glycerol, using an Olympus MVX10 microscope. Oligonucleotide primers used to amplify and clone cDNA for the production of the *ifi30* ISH probe are listed in Supplemantaryemental Table S1.

**Cell sorting and flow cytometry**. Embryos were incubated with a Liberase-Blendzyme 3 (Roche) solution for 90 min at 33 °C, then dissociated and resuspended in 0.9× PBS-1% fetal calf serum, as previously described[57]. We excluded dead cells by staining with SYTOX-red (Life Technologies), DAPI, or DRAQ7 (Thermo Fischer Scientific) stainings. Cell sorting was performed using an Aria II (BD Biosciences, software diva v6.1.3) or a BIORAD S3 cell sorter. For ROS detection, 25–30 embryos (*kdrl:eGFP*) were dissociated in 0.9% phosphate-buffered saline (PBS), 0.2 mg/mL liberase (Roche) for 2 h at 32 °C with agitation. Cell pellets were washed in PBS 0.9% containing 1% Fetal Calf Serum and centrifuged again for 5 mins. The supernatant was removed. Cells were placed in Hank's Balanced Salt Solution (HBSS) with 5 mM CellROX deep red reagent (ThermoFisher) and incubated at room temperature for 30 min. Cell suspension was passed through a 40 mm filter prior to flow cytometry. Data were acquired on a LSR2Fortessa (BD Biosciences, software diva8.0.2) and analyzed with FlowJo.

**Quantitative real-time PCR and analysis**. Total RNA was extracted using RNeasy minikit (Qiagen) and reverse transcribed into cDNA using a Superscript III kit (Invitrogen). Quantitative real-time PCR (qPCR) was performed using KAPA SYBR FAST Universal qPCR Kit (KAPA BIOSYSTEMS) and run on a CFX connect real-time system (Bio Rad). All primers are listed in Supplemantaryemental Table S3. Analysis was performed using an unpaired Student *t* test in GraphPad Prism. Each qPCR experiment was performed using biological triplicates. For experiments using qPCR, each n represents an average of biological triplicates from a single experiment. Experiments were repeated at least three times, and fold-change averages from each experiment were combined.

**Oxidative stress detection in vivo**. Whole-mount staining with the MitoSOX and CellROX (Life Technologies) oxidative stress probes was performed on living zebrafish embryos at 48 hpf, following the methods described[56]. Embryos injected with *ifi30*-MO or control-MO were exposed with 5 μM MitoSOX or CellROX solutions for 30 min at 28 C°, followed by analysis using fluorescence or confocal microscopy.

**Confocal microscopy and immunofluorescence staining**. Transgenic fluorescent embryos were embedded in 1% agarose in a glass-bottom dish. Immuno-fluorescence double staining was performed as described previously[57], with chicken anti-GFP (1:400; Life Technologies) and rabbit anti-phospho-histone 3 (pH3) antibodies (1:250; Abcam). AlexaFluor488-conjugated anti-chicken secondary antibody (1:1000; Life Technologies) and AlexaFluor594-conjugated anti-rabbit

secondary antibody (1:1000; Life Technologies) were subsequently utilized to reveal primary antibodies. Confocal imaging was performed using a Nikon inverted A1r spectral microscope.

**Apoptotic-cell detection**. Apoptotic cells were detected by incubating live embryos in 10 μg/mL Acridine Orange (Cayman Chemical) in E3 for 30 min followed by 3 × 10 min-washes in E3. Embryos were anesthetized with Tricaine, mounted on glass-bottom dishes in 0.7% agarose, and imaged.

Apoptotic cells were also examined by TUNEL assay. Embryos were fixed with 2% paraformaldehyde at 4 °C. After gradual rehydration, the embryos were permeabilized with 25 μg/ml proteinase K for 10 minutes at 28 °C followed by 4% paraformaldehyde, and incubated with 90 μl labeling solution plus 10 μl enzyme solution (In Situ Cell Death Detection Kit, Roche) at 37 °C for 2 h. They were washed three times with PBT for 5 min each, and the images were examined by confocal microscopy.

**Time-lapse imaging and analysis**. For time-lapse imaging, *Tg(kdrl:mCherry; cmyb:GFP)* embryos were anaesthetized with 0.03% Tricaine (Sigma), and embedded in 1% agarose in a 60 mm glass-bottom dish. The embryos were imaged at 28.5 °C with a confocal Nikon inverted A1r spectral microscope. Scanning with 20x water immersion objective, z-stalks were acquired with a step size of 7 μm with an interval of 7 min for 6 h in the control and *ifi30*-morphants starting at ~42 hpf, and an interval of 10 min for 7 h in non-treated and during heptanol treatment starting at ~41 hpf. The analysis of *cmyb:GFP*+ cells in all experiments was performed using IMARIS image software (version 9.7.2). The 3D + t acquisitions were manually oriented to have the CHT horizontal. Then, when required, the presence of a 3D translational drift was corrected by detecting structures that are morphologically stable (therefore not intrinsically moving during the time course). Subsequently, a background subtraction was performed on the GFP channel. Finally, for display purposes, maximum intensity projections were used and the intensity of each channel of the images was set in order to saturate the bottom 1% and the top 0.1% of all pixel values. Movies were recorded using Lanczos' resampling with a kernel of size 3 and quantized over 8 bits per channel afterwards.

**Human fetal liver specimens and immunostaining**. Unstained 4 μm sections on whole charged slides were prepared from paraffin blocks. Double staining was identified using a Ventana Discovery Autostainer. *IFI30/CD117* double immunostaining: for the first sequence, we used the monoclonal rabbit anti-CD117/KIT antibody (Clone YR 145, Cell Marque 117R-16, dilution 1:500). Antigen retrieval was performed by heating slides for 64 min in a EDTA pH 8.0 solution at 95 °C. Slides were incubated with the anti-CD117 antibody for 60 min at 36 °C, and detection was performed using an Amplification Kit (Ventana N°760-080) and subsequently the UltraView Universal DAB detection kit (Ventana N°760-500). For the second sequence, with the polyclonal rabbit anti-GILT/IFI30 antibody (PA5-21533, Thermo Fisher, dilution 1:5000), slides were incubated with the anti-IFI30 antibody for 32 mins at 36 °C. This step was followed by detection using the UltraView Universal APR detection Kit (Ventana N°760-501). IFI30/CD68 double immunostaining: for the first sequence with the polyclonal rabbit anti-GILT/IFI30 antibody (PA5-21533, Thermo Fisher, dilution 1:5000), antigen retrieval was performed by heating slides for 64 mins in an EDTA pH 8.0 solution at 95 °C. Slides were incubated with the anti-IFI30 antibody for 32 min at 36 °C and detection was performed using an UltraView Universal DAB detection Kit (Ventana N°760-500). For the second sequence, the monoclonal mouse anti-CD68 antibody (Clone PG-M1, M0876, dilution 1:100) was incubated for 16 min at 37 °C. This step was followed by detection using an OmniMap-HRP Teal (Ventana N° 760-4310). The images were acquired with a Nikon NIS-Elements Version 5.02 microscope. Analysis was performed manually in Image J by counting the number of positive cells on three different slides (one human fetal liver for each developmental stage).

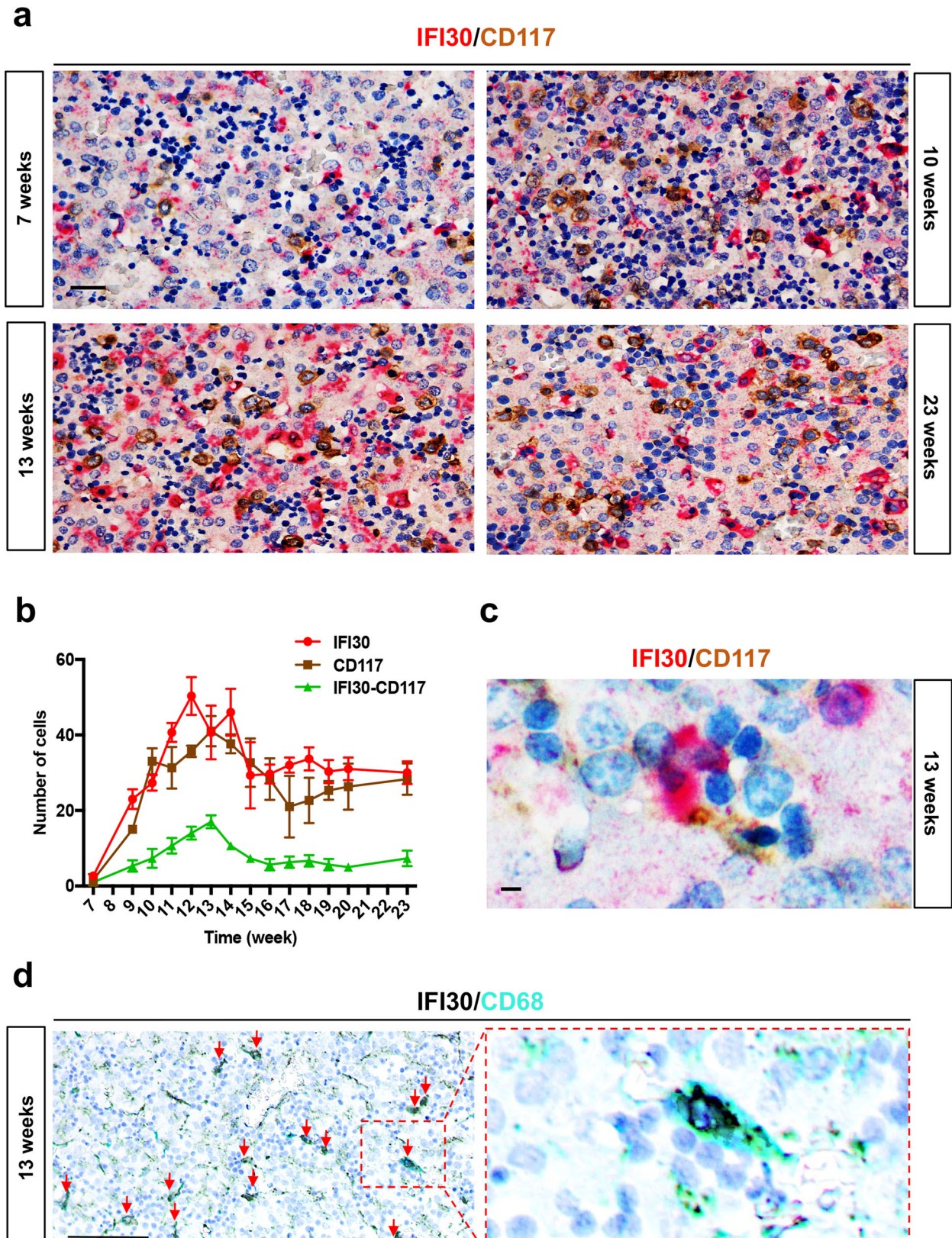

**Fig. 7 HSPCs interact with *IFI30*+ cells in human fetal liver. a** Double immunostaining against *GILT/IFI30* and *CD117* on sections of human fetal liver at different stages of gestation. **b** Distribution of *GILT/IFI30*+ cells (red line), *CD117*+ cells (brown line), and *IFI30*+ cells in contact with *CD117*+ cells (green line). For each developmental stage, only one human fetal liver sample was used ($N = 1$), but three different sections were analyzed. **c** High magnification of double immunostaining against *GILT/IFI30* (red) *CD117* (brown) and nuclei (blue), on sections of human fetal liver at 13 weeks of gestation. Scale bars: 10 μm. **d** Double immunostaining against *GILT/IFI30* (black), *CD68* (cyan), and nuclei (blue) in sections of human fetal liver at 13 weeks of gestation. All red arrows point to double-stained cells. Scale bars: 100 μm (**a**); 25 μm (**c**); 200 μm. Right panel shows a magnification of the region delimited by the red dotted line (**d**).

**Reporting summary**. Further information on research design is available in the Nature Research Reporting Summary linked to this article.

## Data availability

All raw data are freely accessible on the following link: https://doi.org/10.26037/yareta:hxkhluu4szdxnmhegpydld743m, or will be provided by the authors. Source data are provided with this paper.

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

## Acknowledgements

We would like to thank all lab members for their comments. We would also like to thank Pr. B. Kwak for sharing her expertise on connexins, and N. Liaudet from the Bioimaging platform for his help with the Imaris software. J.Y.B. was endorsed by a chair in life sciences funded by the Gabriella Giorgi-Cavaglieri Foundation and is also funded by the Swiss National Fund (grants #31003_166515 and 310030_184814).

## Author contributions

P.C. performed experiments and established the *UAS:ifi30* transgenic line. P.C. and T.P. performed the work on the connexin mutant. C.B.M. performed FACS-sorting cells from *kdrl:GFP* embryos. P.B. and A-L.R. performed immunostainings on human fetal samples. T.P., C.M., and A-L.R. edited the manuscript. P.C. and J.Y.B. designed experiments, performed analysis and wrote the manuscript.

## Competing interests

The authors declare no competing interests.
