## [Peer Review File · Nature Communications]

Reviewers' comments:

Reviewer #1 (Remarks to the Author):

In this manuscript the authors propose a novel mechanism controlling HSC expansion and homeostasis through ROS signaling. Using the zebrafish model, they proposed that expanding HSCs communicate with the vascular niche through ROS release.

The authors major claim is novel and interest to the community, however, the conclusions are not supported by the data. Although an interesting idea, I believe that the story is very preliminary since it is not shown convincingly how and whether ROS move among tissues using gap junctions (if they do). Further experimental evidences are needed to strengthen the conclusions.

There are some unclear points:

- 1) Overall, all confocal analyses of zebrafish CHT region are difficult to follow without some bright field counterstaining. Contrast microscopy should be use to better define the cells involved in these developmental processes.
- 2) H₂O₂ stimulation at 3mM will induce oxidative stress in the whole animal and cardiac function; There are no information about the overall physiology of H₂O₂-treated embryos.
- 3) How do we know that NAC and GSH treatments are working as antioxidants in the zebrafish model (NAC has been shown to produce ROS in specific context: Ezerina et al., 2018) ? ROS levels are very tricky to detect in vivo and ROS signaling vs oxidative stress is challenging to show in zebrafish embryos.
- 4) There is also an overall confusion/misunderstanding among ROS signaling and oxidative stress meanings. The authors use the term "oxidative stress" in the title while they claim ROS production and signaling throughout the text.
- 5) It is reasonable that ROS as H₂O₂ uses aquaporins instead of gap junctions to move across cells.
- 6) The phenotype of the ifi30 mutant is very interesting and should be further investigated.

Reviewer #2 (Remarks to the Author):

In this manuscript, the authors Cacialli et al., using zebrafish as a model, identify a novel pathway that is important for dampening oxidative stress in the caudal hematopoietic tissue (CHT), where HSCs proliferate during development. The authors prove that oxidative stress leads to lower numbers of HSPCs in the CHT and this phenotype can be rescued by co-treating with GSH. They then go on to prove that Ifi30, that is a known lysosomal redox enzyme, plays a role in this process. The authors show that Ifi30 is expressed in endothelial cells of the CHT and its overexpression leads to increased numbers of HSPCs in CHT, but not in AGM. They also perform experiments upon ifi30 knockout and knockdown and show that in the absence of Ifi30 there is a decrease in HSPC proliferation in the CHT. Indeed, absence of Ifi30 leads to accumulation of ROS mostly in endothelial cells and macrophages. Next, they show that ROS travel from HSPCs to endothelial cells, since when they block the connexin channels HSPCs accumulate ROS and eventually die. Finally, the authors show tight association between human HSPCs and IFI30 positive cells in sections from fetal liver. These findings are novel, exciting and could be useful for expanding HSPCs in vitro, but the authors should perform some additional experiments.

Major points

- The authors treat the embryos with hydrogen peroxide and show that HSPC numbers are decreased. What happens to the endothelial cell numbers? Are they unaffected? Or the loss of HSPCs is also due to a decrease of their niche cells? Additionally, can the authors check what happens in the number of endothelial cells when they overexpress ifi30 in these cells? Is it possible that endothelial cells

proliferate more and this is the reason that HSPC numbers increase?

- The authors state that the thiol-reductase activity of Ifi30 depends on two cysteines in the active site. Can the authors check if Ifi30 can rescue the phenotype when these two cysteines are mutated?

- The authors prove that Ifi30 is expressed in endothelial cells by doing qPCR in KDRL+ cells from the tail versus kdrl- cells. Can the authors check ifi30 expression in HSPCs (c-myb+ cells in CHT)?

- Is ifi30 expression affected upon treatment with H₂O₂?

- The authors claim that ROS accumulate in endothelial cells and in macrophages, but not in HSPCs. However, in Supp. Fig 11c and 12a-b it is not obvious that ROS accumulation happens in endothelial cells. The Figs show rather that ROS do not accumulate in HSPCs and accumulate in macrophages. Accumulation in kdrl+ cells should be shown.

- Can the authors do a dose dependent experiment to show that gradual increase in Ifi30 expression rescues HSPCs from oxidative stress? This could be done with ifi30 mRNA upon H₂O₂ treatment.

- Other publications have shown the importance of mitochondrial ROS production for HSPC emergence in the AGM (<https://doi.org/10.1182/blood-2012-12-471201>). How can the authors explain that in their experiments there seems to be no difference in HSPC emergence? Can the authors check ifi30 expression in the endothelial cells of the AGM with qPCR?

Minor comments

- The authors quantify many of their imaging or in situ experiments. However, there are some experiments that lack quantification and the results are not obvious from the one image depicted in the paper. Can the authors quantify the rest of their experiments? For example Supp. Fig. 15a and the whole Supp. Fig. 5 need to be quantified.

- The authors perform some experiments in human fetal liver sections to show "association" of ifi30+ cells with HSPCs. However, the description of these experiments lacks some details. For example, how many sections were tested for each time point, how many embryos were used? Can the authors include these details?

- Can the authors show ROS transfer from HSPCs to endothelial cells with live imaging? That would be very convincing.

- Since there is still space in the main figures, the authors should reduce the numbers of the supplementary figures.

Reviewer #3 (Remarks to the Author):

This paper from the Bertrand group shows a novel role for ifi30 in zebrafish hematopoiesis. ifi30 (gilt) is a lysosomal thiol reductase that is expressed in the zebrafish caudal hematopoietic tissue (CHT), the equivalent of the mammalian fetal liver. ifi30 has been well studied for its role in catalyzing thiol bond reduction and antigen processing. The presented data shows that in ifi30 mutants, ifi30 morphants, and ifi30 over-expression do appear to regulate the number of cmyb+ hematopoietic cells in the embryo. ROS accumulates in the absence of ifi30, and this occurs primarily in macrophages. The authors then investigate the conservation of the role of ifi30 between zebrafish and humans by looking at expression of IFI30 and CD117 in human fetal liver. They show that in human fetal liver, IFI30 expression is limited to macrophages. Overall, the conclusions made are not well supported by the data. In addition, there are many technical limitations to the study that need to be addressed for the conclusions to be convincing.

Major Critiques

- Many of the conclusions made in the paper refer specifically to HSPCs, however cmyb in situ probe is not completely specific to HSPCs. Also, cmyb:gfp transgenic is not a specific reporter line for HSPCs. While cmyb:gfp does mark the small number of HSPCs present in the CHT, it also marks many

different cell types in addition to HSPCs. It is possible that the presented phenotypes are due to a change in number of other populations, such as myeloid cells. The authors should use other HSPC marker lines to confirm the main conclusions of the paper, such as *cd41:gfp* or *runx:mCherry* transgenic lines, and *runx1* in situ probe that are more HSPC-specific. This is particularly relevant in Figure 1A, Fig 2D, and Fig 3A.

- The hydrogen peroxide treatment is high at a concentration of 3 mM. A dose curve is needed to establish the minimal effective dose. How much cell loss is because of general cell death and toxicity? Is it specific to HSPCs? The number of proliferative cells is very low at baseline, so is the decrease consistent with overall decrease in cell number, or are proliferative cells specifically reduced? The GSH rescue effect is very minimal. Was a dose curve performed? Are there other anti-oxidants that could be tested?
- The conclusion that *ifi30* is specifically expressed in ECs is not possible based on the presented in situ data (Supp Fig 2a-b). The resolution of this assay is not sufficient. It could be ECs and hematopoietic or other cells. The qPCR from sorted cells would need to be compared with other specific sorted populations as well, not just bulk of different embryo regions. Single cell RNA-seq of CHT cells (by the authors or using previously published data) may resolve this question more clearly.
- HSPC emergence from the dorsal aorta happens over many hours, so looking at a single timepoint early in the process (Supp Fig 4 and 5, 28 or 36 hpf), does not indicate total production of HSPCs over time, which may be affected by the treatment condition. For example, increase in *rag1* at 4.5 dpf (Supp Fig 6b) could be interpreted as a sustained increase in definitive HSPC production from the *ifi30* over-expression dorsal aorta over time.
- There are multiple assumptions in the conclusion that “over-expression of *ifi30* in the vascular niche can rescue the loss of HSCs after ROS induction in the CHT”. HSPC production could be increased from the dorsal aorta over time. This could be monitored by live imaging. The high level of H₂O₂ may be generally toxic.
- The resolution of the time-lapse CHT movies is very poor and the results cannot be interpreted.
- There needs to be a positive control for ROS induction detected by CellRox—the *ifi30* MO alone is not sufficient. Also the detection reagent needs to be titrated for optimal dose. ROS signal in ECs is not convincing.
- Figure 2 shows an increase in ROS and decrease in HSPCs when *ifi30* is knocked down, while addition of *ifi30* mRNA rescues HSPC number (Suppl, Fig10a-b). Is the rescue of HSPC number due to reduction of ROS after addition of *ifi30* mRNA? Cell-Rox staining on rescued embryos could be performed.
- The specificity of heptanol on gap junctions in these global treatments is not proven, nor is the cell type that is most dependent on functional gap junctions. No more specific investigation into the role of gap junctions by other methods was performed.
- In Supp Fig 15a the number of Mitosox+ cells is very low and not quantified. Is this significant?
- It would appear that the human FL data is consistent with the zebrafish data (Supp Fig. 12), showing that the role of *ifi30* is restricted primarily to macrophages.

Minor Critiques

- Overall the imaging resolution is poor. Larger figure panels, higher resolution and higher magnification images would help.

Reviewers' comments:

Reviewer #1 (Remarks to the Author):

In this manuscript the authors propose a novel mechanism controlling HSC expansion and homeostasis through ROS signaling. Using the zebrafish model, they proposed that expanding HSCs communicate with the vascular niche through ROS release.

The authors major claim is novel and interest to the community, however, the conclusions are not supported by the data. Although an interesting idea, I believe that the story is very preliminary since it is not shown convincingly how and whether ROS move among tissues using gap junctions (if they do). Further experimental evidences are needed to strengthen the conclusions.

There are some unclear points:

Thank you for this evaluation. To clarify, our claim is not that ROS signaling is important between HSCs and their niche, or that the niche controls HSPC expansion through ROS signaling, but the key point is that the niche detoxifies HSCs from their ROS, in order for them to proliferate properly. We have addressed all comments as explained below:

1) Overall, all confocal analyses of zebrafish CHT region are difficult to follow without some bright field counterstaining. Contrast microscopy should be use to better define the cells involved in these developmental processes.

Eventually we have used bright field for counterstaining for some images (Suppl. Fig.10A-B), to define in the CHT, cell-ROX positive cells in *mpeg1:GFP* embryos injected by *ifi30*-morpholino and control-morpholino.

2) H₂O₂ stimulation at 3mM will induce oxidative stress in the whole animal and cardiac function; There are no information about the overall physiology of H₂O₂-treated embryos.

We tested the toxicity of H₂O₂ in the whole animal using several concentrations for 3 hours between 45-48hpf. We identified that 3mM was the best concentration (Suppl.Fig.1A), without any toxicity, and we observed no change in the morphology of embryos (also reported in previous study by Niethammer P, et al., Nature, 2009; Lisse T, et al., Scientific Report, 2016). With 3mM H₂O₂ treatment, we obtained a decrease in the number of *cmv*b expressing-cells in the CHT at 48hpf compared to non-treated embryos (Suppl.Fig.1B).

3) How do we know that NAC and GSH treatments are working as antioxidants in the zebrafish model (NAC has been shown to produce ROS in specific context: Ezerina et al., 2018)? ROS levels are very tricky to detect in vivo and ROS signaling vs oxidative stress is challenging to show in zebrafish embryos.

In the study of Ezerina et al., 2018, the authors used a high concentration of NAC (10mM) in-vitro only. Recent studies in-vivo (Turkmen, R., et al., 2019) showed the antioxidant and cytoprotective effects of N-acetylcysteine (NAC) against the oxidative stress in a rat model. In our study, we employed 10μM (low concentration) for 12 hours. To verify the effect of GSH and NAC as antioxidants, we co-treated with H₂O₂, and we used the cell-rox test to assess ROS content. When we co-treated H₂O₂ with GSH or NAC, the cell-rox signal decreased, a result in favor of the antioxidant role of GSH and NAC (Suppl.Fig.1C).

4) There is also an overall confusion/misunderstanding among ROS signaling and oxidative stress meanings. The authors use the term "oxidative stress" in the title while they claim ROS production and signaling throughout the text.

Thanks for your suggestion. We changed the term as we are not dealing here with ROS signaling, but with ROS detoxification, in order to reduce oxidative stress in HSPCs.

5) It is reasonable that ROS as H₂O₂ uses aquaporins instead of gap junctions to move across cells.

It could have been a possibility, but we have refined our model, based on new results, and now show that ROS are not shuttling between HSCs and their niche, but rather oxidized glutathione. Indeed, we show that HSCs express *gpx1a*, and can therefore process ROS, but are unable to recycle oxidized glutathione (Sup figure 12). This is reinforced by the fact that HSCs can be rescued from connexin inhibition through addition of reduced GSH, which means they can actually perform this step. Therefore, we definitely rule out the possibility that aquaporins are involved

in this process (even though, we have data showing the expression of specific aquaporins by HSCs and their microenvironment).

6) The phenotype of the *ifi30* mutant is very interesting and should be further investigated.

In order to investigate the phenotype of *ifi30* mutants we performed further analysis. First, we evaluated primitive myelopoiesis, performing WISH for *pu1* at 22hpf, and found no difference *ifi30* mutants and their siblings (Suppl.Fig.7A). Next, we evaluated the vascular system and primitive erythropoiesis, performing WISH for *flk1* and *gata1* at 24 and 22hpf, respectively. The expression of these markers was completely normal in *ifi30* mutants comparable to their siblings (Suppl.Fig.7B-C). Further analysis on *ifi30* mutants at 48hpf embryos showed a decrease of *mfap4* (macrophage marker) (Suppl.Fig.7D-E), this result confirms the phenotype that we also observed in *ifi30*-morphants, where macrophages die.

Reviewer #2 (Remarks to the Author):

In this manuscript, the authors Cacialli et al., using zebrafish as a model, identify a novel pathway that is important for dampening oxidative stress in the caudal hematopoietic tissue (CHT), where HSCs proliferate during development. The authors prove that oxidative stress leads to lower numbers of HSPCs in the CHT and this phenotype can be rescued by co-treating with GSH. They then go on to prove that *Ifi30*, that is a known lysosomal redox enzyme, plays a role in this process. The authors show that *Ifi30* is expressed in endothelial cells of the CHT and its overexpression leads to increased numbers of HSPCs in CHT, but not in AGM. They also perform experiments upon *ifi30* knockout and knockdown and show that in the absence of *Ifi30* there is a decrease in HSPC proliferation in the CHT. Indeed, absence of *Ifi30* leads to accumulation of ROS mostly in endothelial cells and macrophages. Next, they show that ROS travel from HSPCs to endothelial cells, since when they block the connexin channels HSPCs accumulate ROS and eventually die. Finally, the authors show tight association between human HSPCs and *IFI30* positive cells in sections from fetal liver. These findings are novel, exciting and could be useful for expanding HSPCs in vitro, but the authors should perform some additional experiments.

We thank reviewer #2 for this positive evaluation, we have tried to address most of your major and minor points.

Major points

- The authors treat the embryos with hydrogen peroxide and show that HSPC numbers are decreased. What happens to the endothelial cell numbers? Are they unaffected? Or the loss of HSPCs is also due to a decrease of their niche cells? Additionally, can the authors check what happens in the number of endothelial cells when they overexpress *ifi30* in these cells? Is it possible that endothelial cells proliferate more and this is the reason that HSPC numbers increase?

To evaluate the impact of H_2O_2 on endothelial cells' numbers, we treated *fli1a:nls-mcherry* transgenic embryos for 3 hours between 45-48hpf with 3mM H_2O_2 . Next, we counted the number of *mcherry*⁺ cells in treated and non-treated embryos, and found no difference (Suppl.Fig.1D). In order to evaluate if the specific overexpression of *ifi30* in endothelial cell increased their number, we outcrossed double transgenic line *kdr1:GAL4;UAS:ifi30* with *fli1a:nuc-mcherry*, and next we screened *fli1-nuc:mcherry;kdr1:GAL4*-positive embryos that were either *UAS:ifi30* negative (upper) or positive (lower) (Suppl.Fig.3F-G). We quantified the number of *mcherry* positive cells in the CHT at 48hpf, and could not observe any difference. Therefore, the *ifi30* loss or gain of function does not impact the number of endothelial cells.

- The authors state that the thiol-reductase activity of *Ifi30* depends on two cysteines in the active site. Can the authors check if *Ifi30* can rescue the phenotype when these two cysteines are mutated?

The *ifi30* mutant presents a stop mutation in the first cysteine, and this event delete the function of the enzyme to reduce disulfide-bond. Previous studies have widely reported that the mutation of either or both cysteine residues abolished reducing activity (Arunachalam B, et al., PNAS 2000; Phan UT, et al., J Biol Chem. 2000; Maric M, et al., Science 2001; for a review see Hastings KT and Cresswell P., Antioxid Redox Signal. 2011). We have not produced a mRNA with a specific mutation at these two codons.

- The authors prove that *Ifi30* is expressed in endothelial cells by doing qPCR in *KDRL*⁺ cells from the tail versus *kdr1*-cells. Can the authors check *ifi30* expression in HSPCs (*c-myb*⁺ cells in CHT)?

We performed qPCR on fluorescence-activated cell sorter (FACS)-sorted by *kdrl*:GFP+, *ikaros*:GFP+ and *mpeg1*:GFP+ cells. We found an enrichment of *ifi30* in endothelial and macrophage cells, but not in HSPCs (Fig.2C), as expected from our in situ hybridizations. Of note, expression of *ifi30* was higher in ECs compared to macrophages.

- Is *ifi30* expression affected upon treatment with H₂O₂?

We performed in situ hybridization for *ifi30* at 48hpf in non-treated and after H₂O₂ treatment. We found no change of *ifi30* expression (Suppl.Fig.4A).

- The authors claim that ROS accumulate in endothelial cells and in macrophages, but not in HSPCs. However, in Supp. Fig 11c and 12a-b it is not obvious that ROS accumulation happens in endothelial cells. The Figs show rather that ROS do not accumulate in HSPCs and accumulate in macrophages. Accumulation in *kdrl*+ cells should be shown.

We managed to use the FACS protocol published by Santoro and colleagues (Mugoni et al, 2014), to show that *kdrl*:GFP+ cells do accumulate ROS (or their derivatives) in the absence of *ifi30* (by morpholino). This result was reproduced 4 independent times and the results are presented on (Figure 4 D-E-F-G). Of note, these experiments are very difficult to achieve, and one should know that even CellRox by itself (without any other treatment) gives a variable background from one experiment to another one.

- Can the authors do a dose dependent experiment to show that gradual increase in *ifi30* expression rescues HSPCs from oxidative stress? This could be done with *ifi30* mRNA upon H₂O₂ treatment.

We performed a dose-dependent experiment injecting *ifi30* mRNA to different concentrations and treating the embryos with H₂O₂. We observe a rescue in the number of HSPCs (Suppl.Fig.4B-C)

- Other publications have shown the importance of mitochondrial ROS production for HSPC emergence in the AGM (<https://doi.org/10.1182/blood-2012-12-471201>). How can the authors explain that in their experiments there seems to be no difference in HSPC emergence? Can the authors check *ifi30* expression in the endothelial cells of the AGM with qPCR?

In Harris et al., Blood 2012, the authors studied the impact of ROS on HSC emergence by treating the embryos during early stages, as early as 12hpf. In our study we focused in a specific time, when HSC colonize the CHT, around 45-48hpf, using H₂O₂ 3mM to induce a significant increase of oxidative stress in this specific short-time.

Minor comments

- The authors quantify many of their imaging or in situ experiments. However, there are some experiments that lack quantification and the results are not obvious from the one image depicted in the paper. Can the authors quantify the rest of their experiments? For example Supp. Fig. 15a and the whole Supp. Fig. 5 need to be quantified.

We have added the quantifications, wherever needed and whenever possible.

- The authors perform some experiments in human fetal liver sections to show “association” of *ifi30*+ cells with HSPCs. However, the description of these experiments lacks some details. For example, how many sections were tested for each time point, how many embryos were used? Can the authors include these details?

We inserted these details in the material and methods section.

- Can the authors show ROS transfer from HSPCs to endothelial cells with live imaging? That would be very convincing.

Unfortunately, that would prove very complicated. Moreover, we now think that oxidized glutathione (GSSG), rather than ROS, are shuttling through connexins. Indeed, HSCs do express *gpx1a*, and are therefore equipped for ROS detoxification, but not for GSSG recycling.

- Since there is still space in the main figures, the authors should reduce the numbers of the supplementary figures.

We reduced the number of figures as much as possible, but added some new supplemental figures to answer reviewers' comments. We now have 7 main figures and 12 supplemental figures.

Reviewer #3 (Remarks to the Author):

This paper from the Bertrand group shows a novel role for *ifi30* in zebrafish hematopoiesis. *ifi30* (*gilt*) is a lysosomal thiol reductase that is expressed in the zebrafish caudal hematopoietic tissue (CHT), the equivalent of the mammalian fetal liver. *ifi30* has been well studied for its role in catalyzing thiol bond reduction and antigen processing. The presented data shows that in *ifi30* mutants, *ifi30* morphants, and *ifi30* over-expression do appear to regulate the number of *cmyb*⁺ hematopoietic cells in the embryo. ROS accumulates in the absence of *ifi30*, and this occurs primarily in macrophages. The authors then investigate the conservation of the role of *ifi30* between zebrafish and humans by looking at expression of *IFI30* and *CD117* in human fetal liver. They show that in human fetal liver, *IFI30* expression is limited to macrophages. Overall, the conclusions made are not well supported by the data. In addition, there are many technical limitations to the study that need to be addressed for the conclusions to be convincing.

Thank you for your evaluation, we have addressed all your points, hopefully in a convincing way.

Major Critiques

- Many of the conclusions made in the paper refer specifically to HSPCs, however *cmyb* in situ probe is not completely specific to HSPCs. Also, *cmyb:gfp* transgenic is not a specific reporter line for HSPCs. While *cmyb:gfp* does mark the small number of HSPCs present in the CHT, it also marks many different cell types in addition to HSPCs. It is possible that the presented phenotypes are due to a change in number of other populations, such as myeloid cells. The authors should use other HSPC marker lines to confirm the main conclusions of the paper, such as *cd41:gfp* or *runx:mCherry* transgenic lines, and *runx1* in situ probe that are more HSPC-specific. This is particularly relevant in Figure 1A, Fig 2D, and Fig 3A.

We agree that *cmyb:eGFP* is not completely specific to HSPCs (although commonly used as such by our community), but it is a very bright transgenic line, therefore very handy to use. Therefore, we performed in situ hybridization for *runx1* (HSPC marker) at 48hpf after injection of control- and *ifi30*-morpholinos, and we also applied GSH treatment for rescue, recapitulating our results (Suppl.Fig.9B-C). To detect the number of HSPCs we also performed an experiment treating Tg(*cd41:GFP*) embryos with heptanol and comparing with non-treated embryos, at 48hpf. We observe a significant decrease of GFP⁺ cells after heptanol treatment (Suppl.Fig.11C). Here again, it should be noted that *CD41:eGFP* is not specific for HSCs, nor HSPCs!

- The hydrogen peroxide treatment is high at a concentration of 3 mM. A dose curve is needed to establish the minimal effective dose. How much cell loss is because of general cell death and toxicity? Is it specific to HSPCs? The number of proliferative cells is very low at baseline, so is the decrease consistent with overall decrease in cell number, or are proliferative cells specifically reduced? The GSH rescue effect is very minimal. Was a dose curve performed? Are there other anti-oxidants that could be tested?

We tested the toxicity of H₂O₂ using several concentrations for 3 hours between around 45-48hpf. We identified that 3mM was the best concentration, not toxic for embryos (also reported in the study of Lisse, T. et al., Scientific Report, 2016), we performed also a survival test for GSH treatment (Suppl.Fig.1A). With H₂O₂ 3 mM treatments, we obtained a decrease in the number of *cmyb* expressing-cells in the CHT at 48hpf compared to non-treated embryos (Suppl.Fig.1B). To verify the effect of GSH and NAC as antioxidants, we co-treated with H₂O₂, and we used the cell-rox test. When we co-treated H₂O₂ and GSH or NAC, the Cell-rox signal decreased, showing the antioxidant properties of these two compounds (Suppl.Fig.1C). We have not tested any other compound.

- The conclusion that *ifi30* is specifically expressed in ECs is not possible based on the presented in situ data (Supp Fig 2a-b). The resolution of this assay is not sufficient. It could be ECs and hematopoietic or other cells. The qPCR from sorted cells would need to be compared with other specific sorted populations as well, not just bulk of different embryo regions. Single cell RNA-seq of CHT cells (by the authors or using previously published data) may resolve this question more clearly.

We performed qPCR on fluorescence-activated cell sorter (FACS)-sorted by *kdr1:GFP*⁺, *ikaros:GFP*⁺ and *mpeg1:GFP*⁺ cells. We found an enrichment of *ifi30* in endothelial and macrophage cells, but not in HSPCs (Fig.2C). The expression in endothelial cells was specific to CHT endothelial cells.

- HSPC emergence from the dorsal aorta happens over many hours, so looking at a single timepoint early in the process (Supp Fig 4 and 5, 28 or 36 hpf), does not indicate total production of HSPCs over time, which may be affected by the treatment condition. For example, increase in rag1 at 4.5 dpf (Supp Fig 6b) could be interpreted as a sustained increase in definitive HSPC production from the ifi30 over-expression dorsal aorta over time.

It might be possible but we did not find an elegant way to test this particular point.

- There are multiple assumptions in the conclusion that “over-expression of ifi30 in the vascular niche can rescue the loss of HSCs after ROS induction in the CHT”. HSPC production could be increased from the dorsal aorta over time. This could be monitored by live imaging. The high level of H₂O₂ may be generally toxic.

We tested the toxicity of H₂O₂ using several concentrations for 3 hours between around 45-48hpf. We identified that 3mM was the best concentration (Suppl.Fig.1A), not toxic for embryos (also reported in the study of Lisse, T. et al., Scientific Report, 2016).

- The resolution of the time-lapse CHT movies is very poor and the results cannot be interpreted.

We processed time lapse movies with the Imaris software to quantify HSPC proliferation in the CHT (Suppl.Fig.7F). To observe the whole CHT at once, we could not use higher magnification, especially as we were already imaging multiple embryos at the same time.

- There needs to be a positive control for ROS induction detected by CellRox—the ifi30 MO alone is not sufficient. Also the detection reagent needs to be titrated for optimal dose. ROS signal in ECs is not convincing.

We have performed quantification of ROS (or derivatives) in endothelial cells by cytometry, using the protocol previously published by Santoro and colleagues (Mugoni, 2014) (Fig. 4D-E-F-G). We have not included all the controls performed for cytometry in the paper, but raw data will be freely accessible on our university server.

- Figure 2 shows an increase in ROS and decrease in HSPCs when ifi30 is knocked down, while addition of ifi30 mRNA rescues HSPC number (Suppl.Fig10a-b). Is the rescue of HSPC number due to reduction of ROS after addition of ifi30 mRNA? Cell-Rox staining on rescued embryos could be performed.

Indeed, we observe a reduction of Cell-Rox staining in ifi30-morphants rescued by ifi30 mRNA (Suppl.Fig.8E).

- The specificity of heptanol on gap junctions in these global treatments is not proven, nor is the cell type that is most dependent on functional gap junctions. No more specific investigation into the role of gap junctions by other methods was performed.

Several studies in vitro and in vivo reported the specificity of Gap junction inhibition by heptanol (Hiroyuki Matsue et al., *J Immunol.* 2006; Brian Bao et al., *FASEB J.* 2010; Tse G, *Mol Med Rep.* 2016; Manjarrez-Marmolejo J and Franco-Pérez J. *Curr Neuropharmacol.* 2016; Yuan, D., et al., *Cell Death Dis.* 2019). Previous studies in zebrafish (Jiang Q, et al., *Fish Physiol Biochem.* 2010) reported a critical role for connexins during HSPCs specification. In particular, cx43-morphants showed a downregulation of hematopoietic stem cell (HSC) marker c-myb in the ventral wall of the dorsal aorta at 36hpf. In the lab, we have several connexin mutants, which could have used, but unfortunately, we observed a similar phenotype with a complete absence of runx1 in the dorsal aorta. In our study we are interested to a specific time during the expansion of HSCs in the CHT at 48hpf. For this reason, we decided to use specific blockers of gap junctions in a defined time, using both heptanol and carbenoxelone. As morpholinos and mutants are not helpful in this particular situation.

- In Supp Fig 15a the number of Mitosox+ cells is very low and not quantified. Is this significant?

We added a new representative figure and the associated quantification in Suppl.Fig.11A.

- It would appear that the human FL data is consistent with the zebrafish data (Supp Fig. 12), showing that the role of ifi30 is restricted primarily to macrophages.

In the zebrafish, we clearly show that ifi30 is expressed in both endothelial cells and macrophages (by qPCR and in situ hybridization). Although it appears that only macrophages express IFI30 in the human fetal liver, it is also possible that we are already too late to observe IFI30 in human fetal liver endothelial cells. Indeed, we are limited in terms of

accessible human stages, by the specimen that are sent to our clinical pathology services. As in zebrafish, where *ifi30* is expressed very transiently in CHT endothelial cells, it is possible that such a pattern of expression could be very dynamic in the human fetal liver.

Minor Critiques

- Overall the imaging resolution is poor. Larger figure panels, higher resolution and higher magnification images would help.

We apologize for that, but we had to include figures at low resolution for the initial submission. The original figures will be higher resolution. And the raw data will be publicly available.

Reviewers' comments:

Reviewer #1 (Remarks to the Author):

To explain the phenotype of zebrafish *ifi30* mutants the authors have now proposed a rather intricate and complex cellular mechanism involving ROS detoxification, ROS shuttling and GSH cycling. The discovery that "HSC expressed Gpx1a and can therefore process ROS, but are unable to recycle oxidized glutathione", as stated by the authors, make the entire model more chaotic and of difficult interpretation. Overall, the conclusions made in this revised manuscript are not supported by the data. There are weaknesses on basic redox biology concepts.

Still some still flaws throughout the manuscript are evident that tone down my enthusiasm for this work:

- The authors did not show that "*ifi30* can recycle oxidase GSSG produced by HSPC into reduced GSH" as stated in the discussion (lines 263-265). Treatment of embryos with l-buthionine sulphoximine (which inhibits gamma-glutamylcysteine synthetase, the enzyme required in the first step of glutathione synthesis and reduces levels of glutathione) would help to understand the role of GSH oxidation and the entire mechanism.
- The authors did not show that "blocking gap junctions by heptanol and carbenoxelone treatment lead to HPSC accumulation of ROS and death by apoptosis" as stated in the discussion (lines 269-270). The chemical treatments (whom specificity I'm doubting, see later) should have been supported by genetic data.
- In vivo KO of connexins (mutants or morpholinos) would be a more convincing way to demonstrate the role of gap junction this process.
- The hypothesis of glutathione transport of back and forth is pure hypothetical and not supported by data and it should be removed.
- While overexpressing *ifi30* increases number of HSPC, and rescues the loss of HSC after a challenge of ROS (e.g. suggesting an antioxidant property), its loss should not decrease HSPC since in basal condition the ROS levels should be physiologically low and do not require antioxidant capacity. It is also weird that H₂O₂ stimulation does not increase *ifi30* expression as expected for an antioxidant enzyme to detoxify from high ROS/oxidative stress. How do the authors explain this phenomena? is there a redox independent function for *ifi30* during development and ROS increase detected by Cell ROX an artifact ?
- To my knowledge such CellROX-positive staining in the CHT ended up in cell death (no senescence). Do they authors measure cell death by caspase3 staining in *ifi30* mutant or morphants ? What are the positive controls for ROS induction detected by Cell Rox ? Do the author Other ROS probes have been tested (e.g. DHE) ?
- Other ROS enhancers such as menadione, BSO, or paraquat have been tested to prove that ROS generation and detoxification are really involved ?
- Heptanol and cabenoxelone has a strong effect on cardiac function and reduced blood flow which might impact on *cmyb* staining (10.1182/blood-2011-05-353235 and 10.1016/j.cell.2009.04.023). This aspect has not been mentioned or discussed.

- It has been shown that ROS produced by mitochondrial activity cause HSC expansion (10.1182/blood-2012-12-471201). In the same paper, NAC treatment show an opposite effect compared to GSH treatments shown here. These is the opposite of the authors are proposing. How these informations cope with these new findings ? The authors did not mention or discuss these previous published data anywhere in the text.
- I found rather misleading and confusing the role of ifi30 in CHT EC and the entire hypothesis of ROS detoxification by the niche.
- Line 188: "ROS did not accumulate in HSPC but rather in ECs (Suppl Fig 9d)": This picture is not showing ROS accumulation in ECs. These data are difficult to understand since the resolution is too low.
- Line 190: "the accumulation of acridine orange caused their senescence (Suppl Fig. 10c-d)": Acridine orange staining does not detect senescence rather autophagy and cell death.

.....

Minor concern:

Fig. 1 : the concentration of H2O2 is too high. Did they check for cell death and toxicity ?

Fig 2a-b: the colocalization with cdh5 is not clear as missing.

Suppl 12a: the colocalization with ikaros is not clear as missing.

Fig 2c: why cmyb line has not been used in these experiments?

Fig 3e: it is not described in the manuscript

Fig 1a,c, Fig 3a, Fig 4c,h, etc: The imaging confocal resolution is poor and clear to understand. Higher resolution, 3D reconstruction and higher magnification images would help.

Reviewer #2 (Remarks to the Author):

The authors have addressed all my comments and have considerably improved their manuscript.

Minor comment: It is great that the authors checked the expression of IFI30 in human fetal livers but I don't think this should be a key point since they only used minimal numbers of fetal livers and have not experimentally proven that the role of IFI30 is similar.

Reviewer #3 (Remarks to the Author):

In this revised version of the manuscript the authors have done extensive additional experiments of high quality to address the individual points raised by the reviewers. However, the data is still over interpreted and model is more complicated than can be explained by the presented data. Rather than providing extensive additional experiments in a second revision that would be beyond the scope of this study, it would be more appropriate to rewrite the text to realistically reflect the data, and save any broader speculation for the discussion section. The data is interesting, but as it is now the conclusions are not supported by the data.

- To address one of the reviewer points, additional data was presented using other HSPC markers that are more specific than cmyb:gfp, such as runx1 in situ probe and cd41:gfp reporter. However, the description of the data in the results section still overstates cmyb:gfp as an HSPC marker. These data

could be described in a more accurate way.

- For the following result the authors should provide some quantification: "Indeed, we observe a reduction of Cell-Rox staining in ifi30-morphants rescued by ifi30 mRNA (Suppl.Fig.8E)." There is no cell counting or indication of how many replicate embryos were used to make this conclusion.
- Specificity of heptanol is still in question. The authors argue that previous studies demonstrate the specificity to target gap junctions, yet treatments of the small molecule to whole embryos could produce effects in any number of tissues.
- For this statement, "We added a new representative figure and the associated quantification in Suppl.Fig.11A", are the authors actually referring to Supp. Fig. 11D? If yes, could the authors show split single channel images and a merge to clearly show the double-positive cells pointed out with arrowheads?
- The authors' state in their rebuttal that, "the key point is that the niche detoxifies HSCs from their ROS, in order for them to proliferate properly." However, the model presented is still overly complicated and the text must clearly and realistically reflect the data.
- It is unclear what this statement means in the rebuttal, as there is no aquaporin data in the paper: "Therefore, we definitely rule out the possibility that aquaporins are involved in this process (even though, we have data showing the expression of specific aquaporins by HSCs and their microenvironment)."
- If a strong claim is made of gap junctions between HSPC and EC, or HSPC and macrophage, this would need to be confirmed with protein localization or EM data. Currently, the role of connexins in their model is inferred from the use of a whole embryo treatment with a small molecule that could have any number of effects that result in increased ROS in HSPCs resulting in their apoptosis. The drug rescue with GSH is also in the whole embryo.
- There is no functional data for gpx1a, and the assumption that it is necessary at this stage of HSPC development is based on its expression pattern.
- One of the "Key Points" states, "In the human fetal liver, IFI30 is expressed by macrophages and plays a similar role", suggesting there is functional data from human fetal liver but there is not. Only expression data is presented.
- Pg.8, line 199 – "by in situ hybridization (Suppl. Fig.11a-b)" should specify that c-myb probe was used

Reviewers' comments:

Reviewer #1 (Remarks to the Author):

To explain the phenotype of zebrafish *ifi30* mutants the authors have now proposed a rather intricate and complex cellular mechanism involving ROS detoxification, ROS shuttling and GSH cycling. The discovery that “HSC expressed Gpx1a and can therefore process ROS, but are unable to recycle oxidized glutathione”, as stated by the authors, make the entire model more chaotic and of difficult interpretation. Overall, the conclusions made in this revised manuscript are not supported by the data. There are weaknesses on basic redox biology concepts.

Still some still flaws throughout the manuscript are evident that tone down my enthusiasm for this work:

- The authors did not show that “*ifi30* can recycle oxidase GSSG produced by HSPC into reduced GSH” as stated in the discussion (lines 263-265). Treatment of embryos with l-buthionine sulphoximine (which inhibits gamma-glutamylcysteine synthetase, the enzyme required in the first step of glutathione synthesis and reduces levels of glutathione) would help to understand the role of GSH oxidation and the entire mechanism.

To evaluate the impact of GSH oxidation on HSCs during expansion in the CHT, we treated embryos with 10µM l-buthionine sulphoximine (BSO) inhibitor of gamma-glutamylcysteine synthetase for 10h (38-48hpf). Next, we performed in situ hybridization for *cmyb* at 48hpf. We found a moderate decrease in the number of *cmyb* expressing cells in the CHT of the embryos treated with BSO (Suppl.Fig.2c-d). We also evaluated the impact of the oxidized glutathione GSSG. We treated the embryos with 10µM GSSG for 10h (38-48hpf), and we performed in situ hybridization for *cmyb* at 48hpf, which resulted in a massive, significant decrease of *cmyb* expressing cells.

These results confirm the involvement of the glutathione pathway during HSC expansion (Suppl.Fig.2e-f).

- The authors did not show that “blocking gap junctions by heptanol and carbenoxelone treatment lead to HPSC accumulation of ROS and death by apoptosis” as stated in the discussion (lines 269-270). The chemical treatments (whom specificity I’m doubting, see later) should have been supported by genetic data.

We added a genetic model with the *connexin41.8^{l11}* knock-out zebrafish. The role of this connexin has been previously described (EMBO Rep. 2006, doi: 10.1038/sj.embor.7400757; eLife 2014 doi.org/10.7554/eLife.05125.001) for pigmentation patterning and recently we reported that this connexin has an important role in the vasculature (Front. Physiol., 2019, doi.org/10.3389/fphys.2019.00080). In zebrafish embryo at 48hpf, we found by qPCR analysis on sorted *kdrl*:GFP+, *ikaros*:GFP+, *mpeg1*:GFP+ cells, that *cx41.8* is enriched in caudal ECs, as well as in

HSPCs and to a lesser extent, macrophages (Fig.5a-b). To investigate the loss of function of this connexin on HSC biology, we performed in situ hybridization for *runx1* and *cmyb* at different stages. We found that *cx41.8*^{-/-} mutants have normal emergence and specification of HSCs (*runx1* at 28hpf, *cmyb* at 36hpf). However, we found that *cx41.8*^{-/-} embryos present a significant decrease of HSCs in the CHT starting at 48hpf, before they are completely lost at 3.5dpf in the CHT. These results confirm previous data in the mouse model, in which the deficiency of connexins induced a defect of HSCs expansion/maintenance in the bone marrow niche (Singh AK, Cancelas JA. *Int J Mol Sci.* 2020). Mechanistically, they showed that the connexins regulate stress-induced hematopoietic regeneration following the transfer of 5-FU-induced ROS from HSPCs to the bone marrow niche, and prevent ROS-p38-p16/INK4a mediated HSPC cell-death (Singh AK, Cancelas JA. *Int J Mol Sci.* 2020). To verify the hypothesis that the connexin-deficiency increases the level of ROS, we detected ROS (by Cell-ROX staining) in *cx41.8*^{-/-} comparing with wild type embryos at 60hpf. We found a significant increase of Cell-Rox positive cells in the CHT of *cx41.8*^{-/-} mutants (Suppl.Fig.12f-g). Next, we tried to rescue the loss of HSCs in the CHT of *cx41.8*^{-/-} mutants by GSH treatment. We treated *cx41.8*^{-/-} embryos for 12 hours (48-60hpf) with GSH 10μM and next measured *cmyb* at 60hpf in the CHT. We found that GSH treatment rescued the number of HSCs in *cx41.8*^{-/-} (Fig.5c-d). These results bring to light an important role of the connexin *cx41.8* during HSC expansion in the CHT.

To verify the effects of Heptanol treatment at 50μM (low concentration) on heart and blood circulation, we performed video time-lapse imaging. We first compared non-treated *myl7:DsRed* embryos (heart marker), with heptanol-treated (50μM, the concentration that we used for our study), and we also tested heptanol 1mM, a high concentration used in previous studies to induce heart pump and blood circulation defects (Commun Integr Biol. 2011doi:10.4161/cib.4.5.15848). We observe in (Suppl.Video 3-4) no defect after treatment with heptanol 50μM. But when we used high concentration of heptanol 1mM, we could reproduce previous studies, with the heart showing pumping defects (Suppl. Video 5). Next, we treated *globin:GFP* embryos (to check blood circulation) with heptanol 50μM and with heptanol 1mM. We observed no difference of blood circulation after heptanol treatment 50μM comparing with non-treated embryos (Suppl. Video 6-7). Embryos treated with Heptanol 1mM showed a defective blood circulation (Suppl.Video 8).

WE CAN CONCLUDE THAT HEPTANOL 50μM, THE CONCENTRATION THAT WE USED IN THIS STUDY DOES NOT PRODUCE ADVERSE EFFECTS IN THE CARDIOVASCULAR TISSUE, THAT IS IMPORTANT FOR HSCs.

Heptanol has been previously used a specific Gap Junction blocker by the group of Koichi Kawakami (Commun Integr Biol. 2011 Sep-Oct; 4(5): 566–568. doi: [10.4161/cib.4.5.15848](https://doi.org/10.4161/cib.4.5.15848)).

- In vivo KO of connexins (mutants or morpholinos) would be a more convincing way to demonstrate the role of gap junction this process.

We added a genetic model with the connexin *41.8*^{ttt1} knock-out zebrafish. The role of this connexin has been previously described (EMBO Rep. 2006, doi:

10.1038/sj.embor.7400757; eLife 2014 doi.org/10.7554/eLife.05125.001) for pigmentation patterning and recently we reported that this connexin has an important role in the vasculature (Front. Physiol., 2019, doi.org/10.3389/fphys.2019.00080). In zebrafish embryo at 48hpf, we found by qPCR analysis on sorted kdrl:GFP+, ikaros:GFP+, mpeg1:GFP+ cells, that cx41.8 is enriched in caudal ECs, as well as in HSPCs and to a lesser extent, macrophages (Fig.5a-b). To investigate the loss of function of this connexin on HSC biology, we performed in situ hybridization for *runx1* and *cmyb* at different stages. We found that *cx41.8*^{-/-} mutants have normal emergence and specification of HSCs (*runx1* at 28hpf, *cmyb* at 36hpf). However, we found that *cx41.8*^{-/-} embryos present a significant decrease of HSCs in the CHT starting at 48hpf, before they are completely lost at 3.5dpf in the CHT. These results confirm previous data in the mouse model, in which the deficiency of connexins induced a defect of HSCs expansion/maintenance in the bone marrow niche (*Singh AK, Cancelas JA. Int J Mol Sci. 2020*). Mechanistically, they showed that the connexins regulate stress-induced hematopoietic regeneration following the transfer of 5-FU-induced ROS from HSPCs to the bone marrow niche, and prevent ROS-p38-p16/INK4a mediated HSPC cell-death (*Singh AK, Cancelas JA. Int J Mol Sci. 2020*). To verify the hypothesis that the connexin-deficiency increases the level of ROS, we detected ROS (by Cell-ROX staining) in *cx41.8*^{-/-} comparing with wild type embryos at 60hpf. We found a significant increase of Cell-Rox positive cells in the CHT of *cx41.8*^{-/-} mutants (Suppl.Fig.12f-g). Next, we tried to rescue the loss of HSCs in the CHT of *cx41.8*^{-/-} mutants by GSH treatment. We treated *cx41.8*^{-/-} embryos for 12 hours (48-60hpf) with GSH 10μM and next measured *cmyb* at 60hpf in the CHT. We found that GSH treatment rescued the number of HSCs in *cx41.8*^{-/-} (Fig.5c-d). These results bring to light an important role of the connexin *cx41.8* during HSC expansion in the CHT.

These results confirm our previous data obtained by inhibition with drug treatment with heptanol and carbenoxelone.

- The hypothesis of glutathione transport of back and forth is pure hypothetical and not supported by data and it should be removed.

We present a new mechanism model in figure 6e. This point is only raised as a hypothesis in the discussion.

- While overexpressing ifi30 increases number of HSPC, and rescues the loss of HSC after a challenge of ROS (e.g. suggesting an antioxidant property), its loss should not decrease HSPC since in basal condition the ROS levels should be physiologically low and do not require antioxidant capacity. It is also weird that H₂O₂ stimulation does not increase ifi30 expression as expected for an antioxidant enzyme to detoxify from high ROS/oxidative stress. How do the authors explain this phenomena? is there a redox independent function for ifi30 during development and ROS increase detected by Cell ROX an artifact?

To verify the antioxidant property of ifi30 we decided to perform qPCR (more accurate to obtain a clear quantification) on zebrafish embryos at 48hpf after different treatments. We first quantified the expression of ifi30 after 3mM H₂O₂ and menadione

10 μ M administration. We observed an increase of ifi30 expression after administration of these ROS enhancers (Suppl.Fig. 4a).

Next, we quantified the expression of ifi30 by qPCR after administration of l-buthionine sulphoximine BSO 10 μ M (which inhibits the gamma-glutamylcysteine synthetase) and after oxidized L-Glutathione 10 μ M (GSSG)(Suppl.Fig. 4b). Here again, we observed a significant increase of ifi30 expression. We confirmed these results also by in situ hybridization for ifi30 after GSSG treatment(Suppl.Fig. 4c).

These results confirm that ifi30 has an important role as thiol reductase, and that its expression is regulated by oxidative stress.

ROS detection was not an artifact as it was shown by many strategies (CellROx stainings and FACS analysis)(Fig.4d-e-f-g).

- To my knowledge such CellROX-positive staining in the CHT ended up in cell death (no senescence). Do they authors measure cell death by caspase3 staining in ifi30 mutant or morphants ? What are the positive controls for ROS induction detected by Cell Rox ? Do the author Other ROS probes have been tested (e.g. DHE) ?

We changed senescence with cell-death, and apologize for this semantic problem.

To detect cell-death, we performed acridine orange staining after injection with control- and ifi30-morpholinos in embryos. We reported the results in Suppl. Fig.11c-d.

When performing CellRox stainings of FACS analysis, H₂O₂ was used as a positive control to enhance ROS (Suppl.Fig.1c). All FACS data will be available in the Yareta data bank.

- Other ROS enhancers such as menadione, BSO, or paraquat have been tested to prove that ROS generation and detoxification are really involved ?

We used as positive control the best ROS enhancer, H₂O₂, as reported in different highly cited former studies (Cell Commun Signal. 2015. doi.org/10.1186/s12964-015-0118-6; Biochem. Biophys Acta. 2016. doi:10.1016/j.bbamcr.2016.09.012; Sci. Report. 2016. doi.org /10.1038/srep20328).

But we also performed experiments with menadione, another ROS enhancers. We found that after 10h of menadione treatment 10 μ M, the number of HSPCs decreased at 48hpf (Suppl.Fig2a-b). Next, we also evaluated the impact of the glutathione synthesis inhibitor 10 μ M buthionine sulfoximine (BSO) and 10 μ M oxidized glutathione GSSG treatments on HSCs in the CHT. We found a significative decrease in the number of cmyb-expressing cells in the CHT of treated embryos (Suppl.Fig.2c-d-e-f). These results confirm that an oxidative stress is deleterious for HSPCs during their expansion in the CHT.

- Heptanol and cabenoxelone has a strong effect on cardiac function and reduced blood flow which might impact on cmyb staining (10.1182/blood-2011-05-353235 and 10.1016/j.cell.2009.04.023). This aspect has not been mentioned or discussed.

To verify the effects of Heptanol treatment at 50 μ M (low concentration) on heart and blood circulation, we performed video time-lapse imaging. We first compared non-treated myl7:DsRed embryos (heart marker), with heptanol-treated (50 μ M, the concentration that we used for our study), and we also tested heptanol 1mM, a high concentration used in previous studies to induce heart pump and blood circulation defects (*Commun Integr Biol.* 2011doi:10.4161/cib.4.5.15848). We observe in (Suppl.Video 3-4) no defect after treatment with heptanol 50 μ M. But when we used high concentration of heptanol 1mM, we could reproduce previous studies, with the heart showing pumping defects (Suppl. Video 5). Next, we treated globin:GFP embryos (to check blood circulation) with heptanol 50 μ M and with heptanol 1mM. We observed no difference of blood circulation after heptanol treatment 50 μ M comparing with non-treated embryos (Suppl. Video 6-7). Embryos treated with Heptanol 1mM showed a defective blood circulation (Suppl.Video 8).

WE CAN CONCLUDE THAT HEPTANOL 50 μ M, THE CONCENTRATION THAT WE USED IN THIS STUDY DOES NOT PRODUCE ADVERSE EFFECTS IN THE CARDIOVASCULAR TISSUE, THAT IS IMPORTANT FOR HSCs.

Heptanol has been previously used a specific Gap Junction blocker by the group of Koichi Kawakami (*Commun Integr Biol.* 2011 Sep-Oct; 4(5): 566–568. doi: [10.4161/cib.4.5.15848](https://doi.org/10.4161/cib.4.5.15848)).

- It has been shown that ROS produced by mitochondrial activity cause HSC expansion (10.1182/blood-2012-12-471201). In the same paper, NAC treatment show an opposite effect compared to GSH treatments shown here. These is the opposite of the authors are proposing. How these informations cope with these new findings? The authors did not mention or discuss these previous published data anywhere in the text.

This study, mentioned by the reviewer, concerns HSC SPECIFICATION IN THE AORTA. HOWEVER, IN OUR MANUSCRIPT, WE ARE LOOKING AT HSC EXPANSION **IN THE CHT**, AT LATER STAGES!!!!

It is highly possible that one needs ROS signaling to promote EHT, while too much ROS later is deleterious for HSCs, a fact which is highly documented and cited in several studies in zebrafish and mammals (*Blood.* 2010; doi:10.1182/blood-2009-05-222711; *Antioxid Redox Signal.* 2014; doi:10.1089/ars.2014.5941; *The Journal of experimental medicine.* 2015; doi:10.1084/jem.20141286). Also, it has been documented that human HSCs expand better in the presence of NAC, a well-known antioxidant (*Blood.* 2014; doi:10.1182/blood-2014-03-559369; *Blood Adv.* 2018; doi:10.1182/bloodadvances.2018024273).

- I found rather misleading and confusing the role of ifi30 in CHT EC and the entire hypothesis of ROS detoxification by the niche.

Gamma-interferon-inducible lysosomal thiol reductase (ifi30) is the only enzyme known to catalyze disulfide bond reduction in the endocytic pathway (Arunachalam B, et al., Proc Natl Acad Sci U S A. 2000). Previous studies have described in detail that ifi30 regulates cellular redox status in vivo and in vitro (Bogunovic et al., J Biol Chem.2008; Maric et al., J Immunol. 2009; Chiang and Maric, Free Radic Biol Med 2011; Rausch MP and Hastings KT. Mol Immunol. 2015).

In our study we confirm this antioxidant activity of ifi30 by qPCR and in situ hybridization (Suppl. Figure 5a-b-c). And in ifi30-deficient embryos we find an increase of ROS levels in ECs and macrophages (Fig. 4c-d-e-f-g; Suppl. Fig. 9e-f; Suppl. Fig 11a-b). We also confirm the antioxidant activity of ifi30, as its overexpression in the whole embryo, or specifically in ECs (kdrl:Gal4/UAS:ifi30) can rescue the loss of HSCs induced by excess of ROS (Suppl. Fig. 5d-e; Fig.3a-b; Suppl. Fig. 14c-d). We also overexpressed ifi30 in HSCs (gata2b:kaTA4/UAS:ifi30), which rescues HSCs after blocking connexin channels (Suppl. Fig. 14e-f). All these experiments confirm once more the antioxidant role of ifi30.

Concerning the interplay between ROS and connexins, our model confirms data from previous studies. In the mouse, HSCs mostly reside in the bone marrow in a quiescent, nonmotile state via adhesion interactions with stromal cells and macrophages. Quiescent, proliferating, and differentiating stem cells have different metabolism, and accordingly different amounts of intracellular reactive oxygen species (ROS). Low levels of ROS, regulated by intrinsic factors such as cell respiration or nicotinamide adenine dinucleotide phosphate-oxidase (NADPH oxidase) activity, or extrinsic factors such as stem cell factor or prostaglandin E2 are required for maintaining stem cell self-renewal. High ROS levels, due to stress and inflammation, induce HSC cell death. (Ludin A., et al., Nature Immunology 2012 doi: 10.1038/ni.2408; Ludin A., et al., Antioxidants & Redox Signaling 2014 doi: 10.1089/ars.2014.5941).

The fate of HSCs is tightly regulated by a combination of cell-intrinsic (transcriptional and epigenetic regulators) and cell-extrinsic factors (soluble growth factors, cytokines, microbial ligands, and adhesive interactions) (Cabezas-Wallscheid et al., Cell Stem Cell 2014). Several studies have demonstrated cell-to-cell interactions between HSCs and the surrounding niche cells (endothelial cells, stromal cells, and osteoblasts), which are essential for HSC localization, maintenance, and differentiation (Butler et al., Cell Stem Cell 2010; Gori et al., Stem Cell Transl. Med. 2017). **A growing body of work has detailed the importance of connexins Gap junction mediated intercellular communication (GJIC) in the regulation of signaling pathways required for HSC survival, proliferation, and fate decisions (Schainovitz, et al., Nature Immunol. 2011).** It has been reported that the deficiency of connexins induces a defect of HSC expansion/maintenance in the bone marrow niche (for a review see Singh AK, Cancelas JA. *Int J Mol Sci.* 2020). Mechanistically, they showed that the connexins regulate stress-induced hematopoietic regeneration following the transfer of 5FU-induced ROS from HSPCs to the bone marrow niche, and prevents ROS-p38-p16/INK4a mediated HSPC cell-death (Singh AK, Cancelas JA. *Int J Mol Sci.* 2020).

- Line 188: “ROS did not accumulate in HSPC but rather in ECs (Suppl Fig 9d)”: This picture is not showing ROS accumulation in ECs. These data are difficult to understand since the resolution is too low.

We have addressed this issue by presenting FACS data that clearly show accumulation of ROS in endothelial cells (Fig. 4 d-e-f-g). This point was raised by all reviewers after the first round of revision and we successfully showed these new results already in the last version.

- Line 190: “the accumulation of acridine orange caused their senescence (Suppl Fig. 10c-d)”: Acridine orange staining does not detect senescence rather autophagy and cell death.

As mentioned earlier, we changed senescence with cell death.

Fig. 1: the concentration of H₂O₂ is too high. Did they check for cell death and toxicity ?

WE PRESENTED A TOXICITY ASSAY IN SUPPL. FIG 1a already in the last version, as demanded by reviewer 1.

This concentration is not too high, as many studies use higher concentrations.

We tested the toxicity of H₂O₂ in the whole animal using several concentrations for 3 hours between around 45-48hpf. We identified that 3mM was the best concentration (Suppl.Fig.1A), not toxic, as we observed no change in the morphology of embryos (also reported in previous study by Niethammer P, et al., Nature, 2009; Lisse T, et al., Scientific Report, 2016).

Fig 2a-b: the colocalization with *cdh5* is not clear as missing.

We performed qPCR for *ifi30* for each specific cell population (Fig.2c)

Fig 3e: it is not described in the manuscript

Lines 162-165.

Then, we challenged *kdr1:Gal4;UAS:ifi30* zebrafish embryos with H₂O₂ 3mM for three hours starting at ~45hpf (Fig.3e-f). As expected, *cmyb* expression decreased in *kdr1:Gal4* single-positive embryos at 48hpf in the CHT, but the HSPCs number was maintained in *kdr1:Gal4;UAS:ifi30* embryos (Fig.3e-f).

Fig 1a,c, Fig 3a, Fig 4c,h, etc: The imaging confocal resolution is poor and clear to understand. Higher resolution, 3D reconstruction and higher magnification images would help.

We explained in our previous answers that the figures are compressed, but the last version of all figures will be in high resolution.

Reviewer #2 (Remarks to the Author):

The authors have addressed all my comments and have considerably improved their manuscript. Minor comment: It is great that the authors checked the expression of IFI30 in human fetal livers but I don't think this should be a key point since they only used minimal numbers of fetal livers and have not experimentally proven that the role of IFI30 is similar.

Thank you for not adding more comments as we replied to all your concerns from the first round of review.

Reviewer #3 (Remarks to the Author):

In this revised version of the manuscript the authors have done extensive additional experiments of high quality to address the individual points raised by the reviewers. However, the data is still over interpreted and model is more complicated than can be explained by the presented data. Rather than providing extensive additional experiments in a second revision that would be beyond the scope of this study, it would be more appropriate to rewrite the text to realistically reflect the data, and save any broader speculation for the discussion section. The data is interesting, but as it is now the conclusions are not supported by the data.

Thank you for your comments on our work. We have carefully addressed all your points as followed:

- To address one of the reviewer points, additional data was presented using other HSPC markers that are more specific than *cmyb:gfp*, such as *runx1* in situ probe and *cd41:gfp* reporter. However, the description of the data in the results section still overstates *cmyb:gfp* as an HSPC marker. These data could be described in a more accurate way.

All studies concerning HSCs in the zebrafish use *cmyb* as a marker of HSCs; see papers from (Cell. 2009; doi:10.1016/j.cell.2009.04.023; Nature. 2010; doi:10.1038/nature08738; Cell. 2014; doi.org/10.1016/j.cell.2014.10.031; Stem Cell Reports. 2017; doi:10.1016/j.stemcr.2017.05.018). To date, there is no evidence that *cmyb* is marking other things than hematopoietic progenitors in the zebrafish model, which is why we do not use HSCs, but rather HSPCs. This reviewer would be so kind to tell us which reference to look after, if there is any available.

- For the following result the authors should provide some quantification: "Indeed, we observe a reduction of Cell-Rox staining in *ifi30*-morphants rescued by *ifi30* mRNA (Suppl.Fig.8E)." There is no cell counting or indication of how many replicate embryos were used to make this conclusion.

We added the quantification of Cell-ROX⁺ for each condition; in this new version, the data is presented in Suppl.Fig.9e-f.

- Specificity of heptanol is still in question. The authors argue that previous studies demonstrate the specificity to target gap junctions, yet treatments of the small molecule to whole embryos could produce effects in any number of tissues.

We have replied to this comment in our last rebuttal. We do short treatments, and there is no way to treat only parts of the embryo since we put the drugs in water.

To verify the effects of Heptanol treatment at 50 μ M (low concentration) on heart and blood circulation, we performed video time-lapse imaging. We first compared non-treated myl7:DsRed embryos (heart marker), with heptanol-treated (50 μ M, the concentration that we used for our study), and we also tested heptanol 1mM, a high concentration used in previous studies to induce heart pump and blood circulation defects (*Commun Integr Biol.* 2011doi:10.4161/cib.4.5.15848). We observe in (Suppl.Video 3-4) no defect after treatment with heptanol 50 μ M. But when we used high concentration of heptanol 1mM, we could reproduce previous studies, with the heart showing pumping defects (Suppl. Video 5). Next, we treated globin:GFP embryos (to check blood circulation) with heptanol 50 μ M and with heptanol 1mM. We observed no difference of blood circulation after heptanol treatment 50 μ M comparing with non-treated embryos (Suppl. Video 6-7). Embryos treated with Heptanol 1mM showed a defective blood circulation (Suppl.Video 8).

WE CAN CONCLUDE THAT HEPTANOL 50 μ M, THE CONCENTRATION THAT WE USED IN THIS STUDY DOES NOT PRODUCE ADVERSE EFFECTS IN THE CARDIOVASCULAR TISSUE, THAT IS IMPORTANT FOR HSCs.

Heptanol has been previously used a specific Gap Junction blocker by the group of Koichi Kawakami (*Commun Integr Biol.* 2011 Sep-Oct; 4(5): 566–568. doi: [10.4161/cib.4.5.15848](https://doi.org/10.4161/cib.4.5.15848)).

- For this statement, “We added a new representative figure and the associated quantification in Suppl.Fig.11A”, are the authors actually referring to Supp. Fig. 11D? If yes, could the authors show split single channel images and a merge to clearly show the double-positive cells pointed out with arrowheads?

We added single channel images and a merge with high magnification, we also indicated with arrows the double positive cells, in this new version the results are now presented in Suppl.Fig.14a-b.

- The authors’ state in their rebuttal that, “the key point is that the niche detoxifies HSCs from their ROS, in order for them to proliferate properly.” However, the model presented is still overly complicated and the text must clearly and realistically reflect the data.

We presented a new version of the model, supported by new results from our connexin mutant.

- It is unclear what this statement means in the rebuttal, as there is no aquaporin data in the paper: “Therefore, we definitely rule out the possibility that aquaporins are involved in this process (even though, we have data showing the expression of specific aquaporins by HSCs and their microenvironment).”

We also have data showing the expression of aquaporins by HSCs and their microenvironment, but with our new model, we suggest that glutathione metabolites, and not directly ROS are transiting through the channel. And GSH/GSSH are not thought to be able to travel through aquaporins. There could be a number of other channels involved, but we clearly show that our Cx41.8 mutant has a specific effect on HSPCs in the CHT.

- If a strong claim is made of gap junctions between HSPC and EC, or HSPC and macrophage, this would need to be confirmed with protein localization or EM data. Currently, the role of connexins in their model is inferred from the use of a whole embryo treatment with a small molecule that could have any number of effects that result in increased ROS in HSPCs resulting in their apoptosis. The drug rescue with GSH is also in the whole embryo.

We added a genetic model with the connexin *cx41.8*^{tt1} knock-out zebrafish. The role of this connexin has been previously described (EMBO Rep. 2006, doi: 10.1038/sj.embor.7400757; eLife 2014 doi.org/10.7554/eLife.05125.001) for pigmentation patterning and recently we reported that this connexin has an important role in the vasculature (Front. Physiol., 2019, doi.org/10.3389/fphys.2019.00080). In zebrafish embryo at 48hpf, we found by qPCR analysis on sorted kdrl:GFP+, ikaros:GFP+, mpeg1:GFP+ cells, that *cx41.8* is enriched in caudal ECs, as well as in HSPCs and to a lesser extent, macrophages (Fig.5a-b). To investigate the loss of function of this connexin on HSC biology, we performed in situ hybridization for *runx1* and *cmyb* at different stages. We found that *cx41.8*^{-/-} mutants have normal emergence and specification of HSCs (*runx1* at 28hpf, *cmyb* at 36hpf). However, we found that *cx41.8*^{-/-} embryos present a significant decrease of HSCs in the CHT starting at 48hpf, before they are completely lost at 3.5dpf in the CHT. These results confirm previous data in the mouse model, in which the deficiency of connexins induced a defect of HSCs expansion/maintenance in the bone marrow niche (Singh AK, Cancelas JA. *Int J Mol Sci.* 2020). Mechanistically, they showed that the connexins regulate stress-induced hematopoietic regeneration following the transfer of 5-FU-induced ROS from HSPCs to the bone marrow niche, and prevent ROS-p38-p16/INK4a mediated HSPC cell-death (Singh AK, Cancelas JA. *Int J Mol Sci.* 2020). To verify the hypothesis that the connexin-deficiency increases the level of ROS, we detected ROS (by Cell-ROX staining) in *cx41.8*^{-/-} comparing with wild type embryos at 60hpf. We found a significant increase of Cell-Rox positive cells in the CHT of *cx41.8*^{-/-} mutants (Suppl.Fig.12f-g). Next, we tried to rescue the loss of HSCs in the CHT of *cx41.8*^{-/-} mutants by GSH treatment. We treated *cx41.8*^{-/-} embryos for 12 hours (48-60hpf) with GSH 10µM and next measured *cmyb* at 60hpf in the CHT. We found that GSH treatment rescued the number of HSCs in *cx41.8*^{-/-} (Fig.5c-d). These results bring to light an important role of the connexin *cx41.8* during HSC expansion in the CHT.

These results confirm our previous data obtained by inhibition with drug treatment with heptanol and carbenoxelone.

Several studies have previously demonstrated by Electron Microscopy cell-to-cell interactions between HSC and the surrounding niche cells (endothelial cells, stromal cells, and osteoblasts), which are essential for HSC localization, maintenance, and differentiation in mouse and zebrafish (Butler et al., Cell Stem Cell 2010; Tamplin O., et al., Cell 2015; Gori et al., Stem Cell Transl. Med. 2017). A growing body of work has detailed the importance of connexins Gap junction mediated intercellular communication (GJIC) in the regulation of signaling pathways required for HSC survival, proliferation, and fate decisions (Schajnovitz, et al., Nature Immunol. 2011). In particular, electron microscopy in the CHT has been performed to show direct interaction between HSPCs and endothelial cells, also known as “cuddling”.(Tamplin, Cell 2015).

- There is no functional data for gpx1a, and the assumption that it is necessary at this stage of HSPC development is based on its expression pattern.

We removed the gpx1a data from this new version, and only mention this expression in the discussion as a data not shown.

- One of the “Key Points” states, “In the human fetal liver, IFI30 is expressed by macrophages and plays a similar role”, suggesting there is functional data from human fetal liver but there is not. Only expression data is presented.

We removed this key point, and added a key point about the Cx41.8.

- Pg.8, line 199 – “by in situ hybridization (Suppl. Fig.11a-b)” should specify that c-myb probe was used

We did.

REVIEWER COMMENTS

Reviewer #1 (Remarks to the Author):

To explain the phenotype of zebrafish *ifi30* mutants the authors have now proposed a rather intricate and complex cellular mechanism involving ROS detoxification, ROS shuttling and GSH cycling. The discovery that "HSC expressed Gpx1a and can therefore process ROS, but are unable to recycle oxidized glutathione", as stated by the authors, make the entire model more chaotic and of difficult interpretation. Overall, the conclusions made in this revised manuscript are not supported by the data. There are weaknesses on basic redox biology concepts.

Still some still flaws throughout the manuscript are evident that tone down my enthusiasm for this work:

- The authors did not show that "*ifi30* can recycle oxidase GSSG produced by HSPC into reduced GSH" as stated in the discussion (lines 263-265). Treatment of embryos with L-buthionine sulphoximine (which inhibits gamma-glutamylcysteine synthetase, the enzyme required in the first step of glutathione synthesis and reduces levels of glutathione) would help to understand the role of GSH oxidation and the entire mechanism.
- The authors did not show that "blocking gap junctions by heptanol and carbenoxelone treatment lead to HPSC accumulation of ROS and death by apoptosis" as stated in the discussion (lines 269-270). The chemical treatments (whom specificity I'm doubting, see later) should have been supported by genetic data.
- In vivo KO of connexins (mutants or morpholinos) would be a more convincing way to demonstrate the role of gap junction this process.
- The hypothesis of glutathione transport of back and forth is pure hypothetical and not supported by data and it should be removed.
- While overexpressing *ifi30* increases number of HSPC, and rescues the loss of HSC after a challenge of ROS (e.g. suggesting an antioxidant property), its loss should not decrease HSPC since in basal condition the ROS levels should be physiologically low and do not require antioxidant capacity. It is also weird that H₂O₂ stimulation does not increase *ifi30* expression as expected for an antioxidant enzyme to detoxify from high ROS/oxidative stress. How do the authors explain this phenomena? is there a redox independent function for *ifi30* during development and ROS increase detected by Cell ROX an artifact ?
- To my knowledge such CellROX-positive staining in the CHT ended up in cell death (no senescence). Do they authors measure cell death by caspase3 staining in *ifi30* mutant or morphants ? What are the positive controls for ROS induction detected by Cell Rox ? Do the author Other ROS probes have been tested (e.g. DHE) ?
- Other ROS enhancers such as menadione, BSO, or paraquat have been tested to prove that ROS generation and detoxification are really involved ?
- Heptanol and cabenoxelone has a strong effect on cardiac function and reduced blood flow which might impact on *cmyb* staining (10.1182/blood-2011-05-353235 and 10.1016/j.cell.2009.04.023). This aspect has not been mentioned or discussed.

- It has been shown that ROS produced by mitochondrial activity cause HSC expansion (10.1182/blood-2012-12-471201). In the same paper, NAC treatment show an opposite effect compared to GSH treatments shown here. These is the opposite of the authors are proposing. How these informations cope with these new findings ? The authors did not mention or discuss these previous published data anywhere in the text.
- I found rather misleading and confusing the role of ifi30 in CHT EC and the entire hypothesis of ROS detoxification by the niche.
- Line 188: "ROS did not accumulate in HSPC but rather in ECs (Suppl Fig 9d)": This picture is not showing ROS accumulation in ECs. These data are difficult to understand since the resolution is too low.
- Line 190: "the accumulation of acridine orange caused their senescence (Suppl Fig. 10c-d)": Acridine orange staining does not detect senescence rather autophagy and cell death.

.....

Minor concern:

Fig. 1 : the concentration of H2O2 is too high. Did they check for cell death and toxicity ?

Fig 2a-b: the colocalization with cdh5 is not clear as missing.

Suppl 12a: the colocalization with ikaros is not clear as missing.

Fig 2c: why cmyb line has not been used in these experiments?

Fig 3e: it is not described in the manuscript

Fig 1a,c, Fig 3a, Fig 4c,h, etc: The imaging confocal resolution is poor and clear to understand. Higher resolution, 3D reconstruction and higher magnification images would help.

Reviewer #2 (Remarks to the Author):

The authors have addressed all my comments and have considerably improved their manuscript.

Minor comment: It is great that the authors checked the expression of IFI30 in human fetal livers but I don't think this should be a key point since they only used minimal numbers of fetal livers and have not experimentally proven that the role of IFI30 is similar.

Reviewer #3 (Remarks to the Author):

In this revised version of the manuscript the authors have done extensive additional experiments of high quality to address the individual points raised by the reviewers. However, the data is still over interpreted and model is more complicated than can be explained by the presented data. Rather than providing extensive additional experiments in a second revision that would be beyond the scope of this study, it would be more appropriate to rewrite the text to realistically reflect the data, and save any broader speculation for the discussion section. The data is interesting, but as it is now the conclusions are not supported by the data.

- To address one of the reviewer points, additional data was presented using other HSPC markers that are more specific than cmyb:gfp, such as runx1 in situ probe and cd41:gfp reporter. However, the description of the data in the results section still overstates cmyb:gfp as an HSPC marker. These data

could be described in a more accurate way.

- For the following result the authors should provide some quantification: "Indeed, we observe a reduction of Cell-Rox staining in ifi30-morphants rescued by ifi30 mRNA (Suppl.Fig.8E)." There is no cell counting or indication of how many replicate embryos were used to make this conclusion.
- Specificity of heptanol is still in question. The authors argue that previous studies demonstrate the specificity to target gap junctions, yet treatments of the small molecule to whole embryos could produce effects in any number of tissues.
- For this statement, "We added a new representative figure and the associated quantification in Suppl.Fig.11A", are the authors actually referring to Supp. Fig. 11D? If yes, could the authors show split single channel images and a merge to clearly show the double-positive cells pointed out with arrowheads?
- The authors' state in their rebuttal that, "the key point is that the niche detoxifies HSCs from their ROS, in order for them to proliferate properly." However, the model presented is still overly complicated and the text must clearly and realistically reflect the data.
- It is unclear what this statement means in the rebuttal, as there is no aquaporin data in the paper: "Therefore, we definitely rule out the possibility that aquaporins are involved in this process (even though, we have data showing the expression of specific aquaporins by HSCs and their microenvironment)."
- If a strong claim is made of gap junctions between HSPC and EC, or HSPC and macrophage, this would need to be confirmed with protein localization or EM data. Currently, the role of connexins in their model is inferred from the use of a whole embryo treatment with a small molecule that could have any number of effects that result in increased ROS in HSPCs resulting in their apoptosis. The drug rescue with GSH is also in the whole embryo.
- There is no functional data for gpx1a, and the assumption that it is necessary at this stage of HSPC development is based on its expression pattern.
- One of the "Key Points" states, "In the human fetal liver, IFI30 is expressed by macrophages and plays a similar role", suggesting there is functional data from human fetal liver but there is not. Only expression data is presented.
- Pg.8, line 199 – "by in situ hybridization (Suppl. Fig.11a-b)" should specify that c-myb probe was used

Reviewers' comments:

Reviewer #1 (Remarks to the Author):

To explain the phenotype of zebrafish *ifi30* mutants the authors have now proposed a rather intricate and complex cellular mechanism involving ROS detoxification, ROS shuttling and GSH cycling. The discovery that “HSC expressed Gpx1a and can therefore process ROS, but are unable to recycle oxidized glutathione”, as stated by the authors, make the entire model more chaotic and of difficult interpretation. Overall, the conclusions made in this revised manuscript are not supported by the data. There are weaknesses on basic redox biology concepts.

Still some still flaws throughout the manuscript are evident that tone down my enthusiasm for this work:

- The authors did not show that “*ifi30* can recycle oxidase GSSG produced by HSPC into reduced GSH” as stated in the discussion (lines 263-265). Treatment of embryos with l-buthionine sulphoximine (which inhibits gamma-glutamylcysteine synthetase, the enzyme required in the first step of glutathione synthesis and reduces levels of glutathione) would help to understand the role of GSH oxidation and the entire mechanism.

Thank you for this suggestion. To evaluate the impact of GSH oxidation on HSCs during expansion in the CHT, we treated embryos with 10µM l-buthionine sulphoximine (BSO), the inhibitor of gamma-glutamylcysteine synthetase for 10h (between 38 and 48hpf). We then performed in situ hybridization for *cmyb* at 48hpf. We found a moderate decrease in the number of *cmyb*-expressing cells in the CHT of these BSO-treated embryos (Suppl.Fig.2c-d). We also evaluated the impact of the oxidized glutathione (GSSG). We treated the embryos with 10µM GSSG for 10h (38-48hpf), and performed in situ hybridization for *cmyb* at 48hpf, which resulted in a massive, significant decrease of *cmyb*-expressing cells.

These results therefore confirm the involvement of the glutathione pathway during HSC expansion (Suppl.Fig.2e-f).

- The authors did not show that “blocking gap junctions by heptanol and carbenoxelone treatment lead to HPSC accumulation of ROS and death by apoptosis” as stated in the discussion (lines 269-270). The chemical treatments (whom specificity I’m doubting, see later) should have been supported by genetic data.

We added a genetic model with the *connexin41.8^{tt1}* knock-out zebrafish. The role of this connexin has been previously described (EMBO Rep. 2006, doi: 10.1038/sj.embor.7400757; eLife 2014 doi.org/10.7554/eLife.05125.001) in pigmentation patterning and we also reported recently that this connexin has an important role in the vasculature (Front. Physiol., 2019, doi.org/10.3389/fphys.2019.00080). In zebrafish embryo at 48hpf, we found by qPCR

analysis on sorted *kdr1:GFP+*, *ikaros:GFP+*, *mpeg1:GFP+* cells, that *cx41.8* is enriched in caudal ECs, as well as in HSPCs and to a lesser extent in macrophages (Fig.5a-b). To investigate the loss of function of this connexin on HSC biology, we performed in situ hybridization for *runx1* and *cmyb* at different stages. We found that *cx41.8^{-/-}* mutants have normal emergence and specification of HSCs (*runx1* at 28hpf, *cmyb* at 36hpf). However, we found that *cx41.8^{-/-}* embryos present a significative decrease of HSCs in the CHT starting at 48hpf, before they are completely lost at 3.5dpf in the CHT. These results support previous data in the mouse model, in which the deficiency of connexins induced a defect of HSCs maintenance in the bone marrow niche (Singh AK, Cancelas JA. *Int J Mol Sci.* 2020). Mechanistically, they showed that the connexins regulate stress-induced hematopoietic regeneration following the transfer of 5-FU-induced ROS from HSPCs to the bone marrow niche, and prevent ROS-p38-p16/INK4a mediated HSPC cell-death (Singh AK, Cancelas JA. *Int J Mol Sci.* 2020). To verify the hypothesis that the connexin-deficiency increases the level of ROS, we detected ROS (by Cell-ROX staining) in *cx41.8^{-/-}* compared to wild-type embryos at 60hpf. We found a significative increase of Cell-Rox positive cells in the CHT of *cx41.8^{-/-}* mutants (Suppl.Fig.12f-g). Next, we tried to rescue the loss of HSCs in the CHT of *cx41.8^{-/-}* mutants by GSH treatment. We treated *cx41.8^{-/-}* embryos for 12 hours (48-60hpf) with GSH 10 μ M and next measured *cmyb* at 60hpf in the CHT. We found that GSH treatment rescued the number of HSCs in *cx41.8^{-/-}* (Fig.5c-d). These results bring to light an important role of the connexin *cx41.8* during HSC expansion in the CHT.

To verify the effects of Heptanol treatment at 50 μ M (low concentration) and 1mM (high concentration) on heart and blood circulation, we performed further video time-lapse imaging.

Video S3.Time-lapse imaging of the CHT in a non-treated globin:GFP embryo (48hpf).

Video S4.Time-lapse imaging of the CHT in a heptanol-treated (50 μ M) globin:GFP embryo (48hpf).

Video S5.Time-lapse imaging of the heart in a non-treated *myl7:DsRed* embryo (48hpf).

Video S6.Time-lapse imaging of the heart in a heptanol-treated (50 μ M) *myl7:DsRed* embryo (48hpf).

Video S7. Time-lapse imaging of the CHT in a heptanol-treated (1mM) globin:GFP embryo (48hpf).

Video S8.Time-lapse imaging of the heart in a heptanol-treated (1mM) *myl7:DsRed* embryo (48hpf)

We first compared non-treated globin:GFP embryos (to check blood circulation) and heptanol-treated (50 μ M, the concentration that we used for our study). We observed no difference in blood circulation between heptanol-treated and non-treated embryos (Suppl. Video 3-4). Next, we compared non-treated *myl7:DsRed* embryos (heart myocardium marker), with heptanol-treated (50 μ M, the concentration that we used for our study), we observe in (Suppl.Video 5-6) no defect after treatment with heptanol 50 μ M.

But when we use a high concentration of heptanol (1mM), we could reproduce previous studies and induce defects in heart pumping and blood circulation (*Commun*

Integr Biol. 2011doi:10.4161/cib.4.5.15848), with the heart showing pumping defects and defective blood circulation (Suppl. Video 7-8).

We can conclude that heptanol 50µm, the concentration that we used in this study does not produce adverse effects in the cardiovascular tissue and in blood circulation, which is important for HSCs.

Heptanol has been previously used as a specific Gap Junction blocker by the group of Koichi Kawakami (*Commun Integr Biol.* 2011 Sep-Oct; 4(5): 566–568. doi: [10.4161/cib.4.5.15848](https://doi.org/10.4161/cib.4.5.15848)).

- In vivo KO of connexins (mutants or morpholinos) would be a more convincing way to demonstrate the role of gap junction this process.

As we reported in previous answer, we added a genetic model with the connexin41.8^{fl/fl} knock-out zebrafish. The role of this connexin has been previously described (*EMBO Rep.* 2006, doi: 10.1038/sj.embor.7400757; *eLife* 2014 doi.org/10.7554/eLife.05125.001) for pigmentation patterning and recently we reported that this connexin has an important role in endothelial cell proliferation (*Front. Physiol.*, 2019, doi.org/10.3389/fphys.2019.00080). In zebrafish embryos at 48hpf, we found by qPCR analysis (on sorted kdrl:GFP+, ikaros:GFP+, mpeg1:GFP+ cells), that cx41.8 is enriched in caudal ECs, as well as in HSPCs and to a lesser extent in macrophages (Fig.5a-b). To investigate the loss of function of this connexin on HSC biology, we performed in situ hybridization for *runx1* and *cmyb* at different stages. We found that *cx41.8*^{-/-} mutants have normal emergence and specification of HSCs (*runx1* at 28hpf, *cmyb* at 36hpf). However, we found that *cx41.8*^{-/-} embryos present a significative decrease of HSCs in the CHT starting at 48hpf, before HSPCs are completely lost at 3.5dpf in the CHT. These results confirm previous data in the mouse model, in which the deficiency of connexins induced a defect of HSC maintenance in the bone marrow niche (*Singh AK, Cancelas JA. Int J Mol Sci.* 2020). Mechanistically, they showed that the connexins regulate stress-induced hematopoietic regeneration following the transfer of 5-FU-induced ROS from HSPCs to the bone marrow niche, and prevent ROS-p38-p16/INK4a mediated HSPC cell-death (*Singh AK, Cancelas JA. Int J Mol Sci.* 2020). To verify the hypothesis that the connexin-deficiency increases the level of ROS, we detected ROS (by Cell-ROX staining) in *cx41.8*^{-/-} compared to wild-type embryos at 60hpf. We found a significative increase of Cell-Rox positive cells in the CHT of *cx41.8*^{-/-} mutants (Suppl.Fig.12f-g). Next, we tried to rescue the loss of HSCs in the CHT of *cx41.8*^{-/-} mutants by GSH treatment. We treated *cx41.8*^{-/-} embryos for 12 hours (48-60hpf) with GSH 10µM and next measured *cmyb* at 60hpf in the CHT. We found that GSH treatment rescued the number of HSCs in *cx41.8*^{-/-} (Fig.5c-d). These results bring to light an important role of the connexin *cx41.8* during HSC expansion in the CHT.

These results confirm our previous data using heptanol and carbenoxelone as gap junctions' inhibitors.

- The hypothesis of glutathione transport of back and forth is pure hypothetical and not supported by data and it should be removed.

We present a new mechanism model in figure 6e. This point is only raised as a hypothesis in the discussion. We have shown here that the glutathione pathway is important during HSC expansion. Ifi30 acts as an antioxidant as it can recycle GSSG into GSH (our data and Chiang and Maric, *Free Radic Biol Med.* 2011. Aug 1; 51(3):688-99. doi: 10.1016/j.freeradbiomed.2011.05.015.). We show multiple evidence that ifi30 is expressed in endothelial cells, therefore involving a shuttling (through connexins) between the niche and HSPCs.

- While overexpressing ifi30 increases number of HSPC, and rescues the loss of HSC after a challenge of ROS (e.g. suggesting an antioxidant property), its loss should not decrease HSPC since in basal condition the ROS levels should be physiologically low and do not require antioxidant capacity. It is also weird that H₂O₂ stimulation does not increase ifi30 expression as expected for an antioxidant enzyme to detoxify from high ROS/oxidative stress. How do the authors explain this phenomena? is there a redox independent function for ifi30 during development and ROS increase detected by Cell ROX an artifact?

Low levels of ROS, regulated by intrinsic factors such as cell respiration or nicotinamide adenine dinucleotide phosphate-oxidase (NADPH oxidase) activity, or extrinsic factors such as stem cell factor or prostaglandin E2 are required for maintaining stem cell self-renewal. High ROS levels, due to stress and inflammation, induce HSC cell death. (Ludin A., et al., *Nature Immunology* 2012 doi: 10.1038/ni.2408; Ludin A., et al., *Antioxidants & Redox Signaling* 2014 doi: 10.1089/ars.2014.5941).

The loss of ifi30 increases the level of ROS in ECs and macrophages (Fig. 4c-d-e-f-g; Suppl. Fig. 9e-f; Suppl. Fig 11a-b). This probably means that in basal conditions, the levels of ROS are very low thanks to the action of ifi30. We can even go further and consider that ifi30 is one of the major antioxidant molecules in the hematopoietic niche, since its loss is not compensated by other mechanisms and lead to high ROS levels (that we could measure by Cell Rox, by direct imaging or by cytometry)

To verify the antioxidant property of ifi30 we decided to perform qPCR (more accurate to obtain a clear quantification) on zebrafish embryos at 48hpf after different treatments. We first quantified the expression of ifi30 after H₂O₂ and menadione administration. We observed an increase of ifi30 expression after administration of these ROS enhancers (Suppl.Fig. 5a).

Next, we quantified the expression of ifi30 by qPCR after administration of L-buthionine sulphoximine BSO 10µM (which inhibits the gamma-glutamylcysteine synthetase) and after oxidized L-Glutathione 10µM (GSSG)(Suppl.Fig. 5b). Here again, we observed a significant increase of ifi30 expression. We confirmed these results also by in situ hybridization for ifi30 after GSSG treatment, which showed the highest impact on ifi30 expression (Suppl.Fig. 5c).

We also confirm the antioxidant activity of ifi30, as its overexpression in the whole embryo, or specifically in ECs (*kdrl:Gal4/UAS:ifi30*) can rescue the loss of HSCs induced by excess of ROS (Fig.3d-e; Suppl. Fig. 5d-e). We also overexpressed ifi30 directly in HSCs (*gata2b:kalTA4/UAS:ifi30*), which rescued HSCs after blocking

connexin channels (Suppl. Fig. 14e-f). All these experiments confirm once more the antioxidant role of ifi30.

These results confirm that ifi30 has an important role as thiol reductase, and that its expression is regulated by oxidative stress.

We do not consider that ROS detection was an artifact as it was shown by different independent strategies (CellROX stainings and FACS analysis)(Fig.4d-e-f-g).

- To my knowledge such CellROX-positive staining in the CHT ended up in cell death (no senescence). Do they authors measure cell death by caspase3 staining in ifi30 mutant or morphants ? What are the positive controls for ROS induction detected by Cell Rox ? Do the author Other ROS probes have been tested (e.g. DHE) ?

We changed senescence with cell-death, and apologize for this semantic problem.

To detect cell-death, we performed acridine orange staining after injection with control- and ifi30-morpholinos in embryos. We reported the results in Suppl. Fig.11c-d.

When performing CellRox stainings and FACS analysis, H₂O₂ was used as a positive control to enhance ROS. We also used H₂O₂ as control for in-vivo staining (Supp.Fig1c). All FACS data are already available in the Yareta databank, following the link provided in the manuscript (raw data repository).

- Other ROS enhancers such as menadione, BSO, or paraquat have been tested to prove that ROS generation and detoxification are really involved ?

In all our experiments, we always used H₂O₂ as our first positive control as ROS enhancer. This ROS enhancer has been widely used in different highly cited former studies (Cell Commun Signal. 2015. doi.org/10.1186/s12964-015-0118-6; Biochem. Biophys Acta. 2016. doi:10.1016/j.bbamcr.2016.09.012; Sci. Report. 2016. doi.org /10.1038/srep20328).

But we also performed, as suggested by the reviewer, experiments with menadione, another ROS enhancer. We found that after 10h of menadione treatment 10µM, the number of HSPCs decreased at 48hpf (Suppl.Fig2a-b).

Next, we also evaluated the impact of the glutathione synthesis inhibitor 10µM buthionine sulfoximine (BSO) and 10µM oxidized glutathione GSSG treatments on HSCs in the CHT. We found a significative decrease in the number of cmyb-expressing cells in the CHT of treated embryos (Suppl.Fig.2c-d-e-f). These results confirm that oxidative stress is deleterious for HSPCs during their expansion in the CHT.

- Heptanol and cabenoxelone has a strong effect on cardiac function and reduced blood flow which might impact on cmyb staining (10.1182/blood-2011-05-353235 and 10.1016/j.cell.2009.04.023). This aspect has not been mentioned or discussed.

To verify the effects of Heptanol treatment at 50 μ M (low concentration) and 1mM (high concentration) on heart and blood circulation, we performed further video time-lapse imaging.

Video S3. Time-lapse imaging of the CHT in a non-treated globin:GFP embryo (48hpf).

Video S4. Time-lapse imaging of the CHT in a heptanol-treated (50 μ M) globin:GFP embryo (48hpf).

Video S5. Time-lapse imaging of the heart in a non-treated myl7:DsRed embryo (48hpf).

Video S6. Time-lapse imaging of the heart in a heptanol-treated (50 μ M) myl7:DsRed embryo (48hpf).

Video S7. Time-lapse imaging of the CHT in a heptanol-treated (1mM) globin:GFP embryo (48hpf).

Video S8. Time-lapse imaging of the heart in a heptanol-treated (1mM) myl7:DsRed embryo (48hpf)

We first compared non-treated globin:GFP embryos (to check blood circulation) and heptanol-treated (50 μ M, the concentration that we used for our study). We observed no difference in blood circulation between heptanol-treated and non-treated embryos (Suppl. Video 3-4). Next, we compared non-treated myl7:DsRed embryos (heart myocardium marker), with heptanol-treated (50 μ M, the concentration that we used for our study), we observe in (Suppl. Video 5-6) no defect after treatment with heptanol 50 μ M.

But when we use a high concentration of heptanol (1mM), we could reproduce previous studies and induce defects in heart pumping and blood circulation (*Commun Integr Biol.* 2011doi:10.4161/cib.4.5.15848), with the heart showing pumping defects and defective blood circulation (Suppl. Video 7-8).

We can conclude that heptanol 50 μ M, the concentration that we used in this study does not produce adverse effects in the cardiovascular tissue and in blood circulation, which is important for HSCs.

For information, heptanol has been previously used as a specific Gap Junction blocker by the group of Koichi Kawakami (*Commun Integr Biol.* 2011 Sep-Oct; 4(5): 566–568. doi: [10.4161/cib.4.5.15848](https://doi.org/10.4161/cib.4.5.15848)).

- It has been shown that ROS produced by mitochondrial activity cause HSC expansion (10.1182/blood-2012-12-471201). In the same paper, NAC treatment show an opposite effect compared to GSH treatments shown here. These is the opposite of the authors are proposing. How these informations cope with these new findings? The authors did not mention or discuss these previous published data anywhere in the text.

This study (10.1182/blood-2012-12-471201), mentioned by the reviewer, concerns HSC specification in the aorta. However, in our manuscript, we are looking at HSC

expansion in the CHT, at later stages. Indeed, in our study we tested ROS enhancer and/or antioxidants (GSH and NAC) after 40hpf.

Antioxidants treatments (with NAC) after 40hpf, as reported in a previous study, increased *cmyb* expression in the CHT (Rissone et al. *The Journal of experimental medicine*. 2015; doi:10.1084/jem.20141286). Also in our study, we have shown that the overexpression and/or loss of *ifi30* does not affect HSC specification in the aorta. But we found that *ifi30* plays an important role during HSC expansion in the CHT.

Low levels of ROS, regulated by intrinsic factors such as cell respiration or NADPH oxidase activity, or extrinsic factors such as stem cell factor or prostaglandin E2 are required for maintaining stem cell self-renewal. High ROS levels, due to stress and inflammation, induce HSC cell death in bone marrow and in the CHT niche. All this is highly documented and cited in several studies in zebrafish and mammals (*The Journal of experimental medicine*. 2015; doi:10.1084/jem.20141286; *Blood*. 2010; doi:10.1182/blood-2009-05-222711; Ludin A., et al., *Nature Immunology* 2012 doi: 10.1038/ni.2408; *Antioxid Redox Signal*. 2014; doi:10.1089/ars.2014.5941). Also, it has been documented that human HSCs expand better in the presence of NAC, a well-known antioxidant (*Blood*. 2014; doi:10.1182/blood-2014-03-559369 ; *Blood Adv*. 2018; doi:10.1182/bloodadvances.2018024273).

- I found rather misleading and confusing the role of *ifi30* in CHT EC and the entire hypothesis of ROS detoxification by the niche.

While we appreciate the reviewer's feedback, we respectfully disagree. Gamma-interferon-inducible lysosomal thiol reductase (*ifi30*) is the only enzyme known to catalyze disulfide bond reduction in the endocytic pathway (Arunachalam B, et al., *Proc Natl Acad Sci U S A*. 2000). Previous studies have described in detail that *ifi30* regulates cellular redox status in vivo and in vitro (Bogunovic et al., *J Biol Chem*.2008; Maric et al., *J Immunol*. 2009; Chiang and Maric, *Free Radic Biol Med* 2011; Rausch MP and Hastings KT. *Mol Immunol*. 2015).

In our study, we confirm this antioxidant activity of *ifi30* by qPCR and in situ hybridization (Suppl. Figure 5a-b-c), as its expression is augmented after oxidative stress. Moreover in *ifi30*-deficient embryos, we find an increase of ROS levels in ECs and macrophages (Fig. 4c-d-e-f-g; Suppl. Fig. 9e-f; Suppl. Fig 11a-b). We also confirm the antioxidant activity of *ifi30*, as its overexpression in the whole embryo, or specifically in ECs (*kdr1:Gal4/UAS:ifi30*) can rescue the loss of HSCs induced by the excess of ROS (Suppl. Fig. 5d-e;Fig.3d-e). We also overexpressed *ifi30* in HSCs (*gata2b:ka1TA4/UAS:ifi30*), which rescues HSCs after blocking connexin channels (Suppl. Fig. 14e-f), which we show concentrates ROS in HSPCs themselves. All these experiments confirm once more the antioxidant role of *ifi30*.

Concerning the interplay between ROS and connexins, our model confirms data from previous studies. In the mouse, HSCs mostly reside in the bone marrow in a quiescent, nonmotile state via adhesion interactions with stromal cells and macrophages. Quiescent, proliferating, and differentiating stem cells have different metabolism, and accordingly different amounts of intracellular reactive oxygen species (ROS). Low levels of ROS, regulated by intrinsic factors such as cell

respiration or nicotinamide adenine dinucleotide phosphate-oxidase (NADPH oxidase) activity, or extrinsic factors such as stem cell factor or prostaglandin E2 are required for maintaining stem cell self-renewal. High ROS levels, due to stress and inflammation, induce HSC cell death. (Ludin A., et al., Nature Immunology 2012 doi: 10.1038/ni.2408; Ludin A., et al., Antioxidants & Redox Signaling 2014 doi: 10.1089/ars.2014.5941).

The fate of HSCs is tightly regulated by a combination of cell-intrinsic (transcriptional and epigenetic regulators) and cell-extrinsic factors (soluble growth factors, cytokines, microbial ligands, and adhesive interactions) (Cabezas-Wallscheid et al., Cell Stem Cell 2014). Several studies have demonstrated cell-to-cell interactions between HSCs and the surrounding niche cells (endothelial cells, stromal cells, and osteoblasts), which are essential for HSC localization, maintenance, and differentiation (Butler et al., Cell Stem Cell 2010; Gori et al., Stem Cell Transl. Med. 2017).

A growing body of work has detailed the importance of connexins Gap junction mediated intercellular communication (GJIC) in the regulation of signaling pathways required for HSC survival, proliferation, and fate decisions (Schajnovitz, et al., Nature Immunol. 2011).

It has been reported that the deficiency of connexins induces a defect of HSC expansion/maintenance in the bone marrow niche (for a review see *Singh AK, Cancelas JA. Int J Mol Sci. 2020*). Mechanistically, they showed that the connexins regulate stress-induced hematopoietic regeneration following the transfer of 5FU-induced ROS from HSPCs to the bone marrow niche, and prevents ROS-p38-p16/INK4a mediated HSPC cell-death (*Singh AK, Cancelas JA. Int J Mol Sci. 2020*).

- Line 188: “ROS did not accumulate in HSPC but rather in ECs (Suppl Fig 9d)”: This picture is not showing ROS accumulation in ECs. These data are difficult to understand since the resolution is too low.

We have addressed this issue by presenting FACS data that clearly show accumulation of ROS in endothelial cells (kdrl-positive) (Fig. 4 d-e-f-g). This point was raised by all reviewers after the first round of revision and we successfully showed these new results already in the last version. This data reinforces our live imaging, which showed specific augmentation of ROS in the CHT.

- Line 190: “the accumulation of acridine orange caused their senescence (Suppl Fig. 10c-d)”: Acridine orange staining does not detect senescence rather autophagy and cell death.

As mentioned earlier, we changed senescence with cell death.

Fig. 1: the concentration of H2O2 is too high. Did they check for cell death and toxicity ?

We have presented a toxicity assay in Suppl. Fig 1a.

This concentration is not too high, as many studies use higher concentrations.

We tested the toxicity of H₂O₂ in the whole animal using several concentrations for 3 hours between around 45-48hpf. We identified that 3mM was the best concentration (Suppl.Fig.1A), not toxic, as we observed no change in the morphology of embryos (also reported in previous study by Niethammer P, et al., Nature, 2009; Lisse T, et al., Scientific Report, 2016).

Fig 2a-b: the colocalization with *cdh5* is not clear as missing.

To reinforce our double in situ data, we performed qPCR for *ifi30* for each cell population that we could sort in the CHT and showed that *ifi30* was highly enriched in CHT-ECs (Fig.2c).

Fig 3e: it is not described in the manuscript

The data from Fig3e is described on page 7 in the last paragraph. "Next, we challenged *kdr1:Gal4;UAS:ifi30* zebrafish embryos with H₂O₂ 3mM for three hours starting at ~45hpf (Fig.3e-f). As expected, *cmyb* expression decreased in *kdr1:Gal4* single-positive embryos at 48hpf in the CHT, but the HSPCs number was maintained in *kdr1:Gal4;UAS:ifi30* embryos (Fig.3e-f)".

Fig 1a,c, Fig 3a, Fig 4c,h, etc: The imaging confocal resolution is poor and clear to understand. Higher resolution, 3D reconstruction and higher magnification images would help.

We explained in our previous answers that the figures are compressed, but the last version of all figures will be in high resolution. For this present version, we have augmented the resolution to its maximum, while trying to keep the weight of our figures to its minimum.

Suppl 12a: the colocalization with *ikaros* is not clear as missing.

We have removed this data, which was not supported by functional evidence at the time, and now present a new version of the model, supported by new results from our connexin mutant.

Fig 2c: why *cmyb* line has not been used in these experiments?

For our qPCR experiments, we did not use the *cmyb*:GFP reporter line because it is also expressed in neuronal cells. For these experiments, we used *ikaros*:GFP that is widely used as an HSPC marker by different labs in our community (Zon, North, Feng Liu, Paw, Ward), as reported in previous studies (Cooney, J.D. et al., Developmental Biology 2013, doi: 10.1016/j.ydbio.2012.08.015; Mommaerts H, et al., Developmental Dynamics 2014 doi: 10.1002/dvdy.24164; Wang S, et al., Developmental Cell 2015, doi: 10.1016/j.devcel.2015.07.011). We have previously shown that *ikaros*:GFP positive cells represent HSPCs (Mahony, Stem Cell Reports, 2018).

Reviewer #2 (Remarks to the Author):

The authors have addressed all my comments and have considerably improved their manuscript. Minor comment: It is great that the authors checked the expression of IFI30 in human fetal livers but I don't think this should be a key point since they only used minimal numbers of fetal livers and have not experimentally proven that the role of IFI30 is similar.

We would like to thank this reviewer for not adding more comments as we replied to all his concerns from the first round of review.

Reviewer #3 (Remarks to the Author):

In this revised version of the manuscript the authors have done extensive additional experiments of high quality to address the individual points raised by the reviewers. However, the data is still over interpreted and model is more complicated than can be explained by the presented data. Rather than providing extensive additional experiments in a second revision that would be beyond the scope of this study, it would be more appropriate to rewrite the text to realistically reflect the data, and save any broader speculation for the discussion section. The data is interesting, but as it is now the conclusions are not supported by the data.

Thank you for your comments on our work. We have carefully addressed all your points as followed:

- To address one of the reviewer points, additional data was presented using other HSPC markers that are more specific than *cmyb:gfp*, such as *runx1* in situ probe and *cd41:gfp* reporter. However, the description of the data in the results section still overstates *cmyb:gfp* as an HSPC marker. These data could be described in a more accurate way.

All studies concerning HSCs in the zebrafish use *cmyb* as a marker of HSPCs; see papers from (Cell. 2009; doi:10.1016/j.cell.2009.04.023; Nature. 2010; doi:10.1038/nature08738; Cell. 2014; doi.org/10.1016/j.cell.2014.10.031; Stem Cell Reports. 2017; doi:10.1016/j.stemcr.2017.05.018). To date, there is no evidence that *cmyb* is marking other things than hematopoietic progenitors in the zebrafish model, which is why we do not use HSCs, but rather HSPCs. However, we would be happy to hear about which reference to look after, if there is any available.

- For the following result the authors should provide some quantification: "Indeed, we observe a reduction of Cell-Rox staining in *ifi30*-morphants rescued by *ifi30* mRNA (Suppl.Fig.8E)." There is no cell counting or indication of how many replicate embryos were used to make this conclusion.

We added the quantification of Cell-ROX⁺ for each condition; in this new version, the data is presented in Suppl.Fig.9e-f. For each condition, 13 to 16 embryos have been used, and we carefully counted the number of cell-rox positive cells per CHT.

- Specificity of heptanol is still in question. The authors argue that previous studies demonstrate the specificity to target gap junctions, yet treatments of the small molecule to whole embryos could produce effects in any number of tissues.

To verify the effects of Heptanol treatment at 50µM (low concentration) and 1mM (high concentration) on heart and blood circulation, we performed further video time-lapse imaging.

Video S3. Time-lapse imaging of the CHT in a non-treated globin:GFP embryo (48hpf).

Video S4. Time-lapse imaging of the CHT in a heptanol-treated (50µM) globin:GFP embryo (48hpf).

Video S5. Time-lapse imaging of the heart in a non-treated myl7:DsRed embryo (48hpf).

Video S6. Time-lapse imaging of the heart in a heptanol-treated (50µM) myl7:DsRed embryo (48hpf).

Video S7. Time-lapse imaging of the CHT in a heptanol-treated (1mM) globin:GFP embryo (48hpf).

Video S8. Time-lapse imaging of the heart in a heptanol-treated (1mM) myl7:DsRed embryo (48hpf)

We first compared non-treated globin:GFP embryos (to check blood circulation) and heptanol-treated (50µM, the concentration that we used for our study). We observed no difference in blood circulation between heptanol-treated and non-treated embryos (Suppl. Video 3-4). Next, we compared non-treated myl7:DsRed embryos (heart myocardium marker), with heptanol-treated (50µM, the concentration that we used for our study), we observe in (Suppl.Video 5-6) no defect after treatment with heptanol 50µM.

But when we use a high concentration of heptanol (1mM), we could reproduce previous studies and induce defects in heart pumping and blood circulation (*Commun Integr Biol.* 2011doi:10.4161/cib.4.5.15848), with the heart showing pumping defects and defective blood circulation (Suppl. Video 7-8).

We can conclude that heptanol 50µM, the concentration that we used in this study does not produce adverse effects in the cardiovascular tissue and in blood circulation, which is important for HSCs.

For information, heptanol has been previously used as a specific Gap Junction blocker by the group of Koichi Kawakami (*Commun Integr Biol.* 2011 Sep-Oct; 4(5): 566–568. doi: [10.4161/cib.4.5.15848](https://doi.org/10.4161/cib.4.5.15848)).

- For this statement, “We added a new representative figure and the associated quantification in Suppl.Fig.11A”, are the authors actually referring to Supp. Fig. 11D?

If yes, could the authors show split single channel images and a merge to clearly show the double-positive cells pointed out with arrowheads?

We added single channel images and a merge with high magnification, we also indicated with arrows the double positive cells, in this new version the results are now presented in Suppl.Fig.14a-b.

- The authors' state in their rebuttal that, "the key point is that the niche detoxifies HSCs from their ROS, in order for them to proliferate properly." However, the model presented is still overly complicated and the text must clearly and realistically reflect the data.

We presented a new version of the model, supported by new results from our connexin mutant.

- It is unclear what this statement means in the rebuttal, as there is no aquaporin data in the paper: "Therefore, we definitely rule out the possibility that aquaporins are involved in this process (even though, we have data showing the expression of specific aquaporins by HSCs and their microenvironment)."

We have chosen to present new data using the connexin41.8 mutants. We show here that these mutants have normal HSPC specification from the dorsal aorta, that HSPC migrate normally to the CHT, where they start to be lost. Indeed, by 3.5dpf, HSPCs are completely lost in the CHT, showing important defects in definitive hematopoiesis. We suggest that glutathione metabolites, and not directly ROS are transiting through the channels. GSH and/or GSSH are not thought to be able to travel through aquaporins. There could be a number of other channels involved, but we clearly show that our Cx41.8 mutant has a specific effect on HSPCs in the CHT (see next point for further details).

- If a strong claim is made of gap junctions between HSPC and EC, or HSPC and macrophage, this would need to be confirmed with protein localization or EM data. Currently, the role of connexins in their model is inferred from the use of a whole embryo treatment with a small molecule that could have any number of effects that result in increased ROS in HSPCs resulting in their apoptosis. The drug rescue with GSH is also in the whole embryo.

We added a genetic model with the connexin41.8^{tt1} knock-out zebrafish. The role of this connexin has been previously described (EMBO Rep. 2006, doi: 10.1038/sj.embor.7400757; eLife 2014 doi.org/10.7554/eLife.05125.001) for pigmentation patterning and recently we reported that this connexin has an important role in the vasculature (Front. Physiol., 2019, doi.org/10.3389/fphys.2019.00080). In zebrafish embryo at 48hpf, we found by qPCR analysis (on sorted kdrl:GFP+, ikaros:GFP+, mpeg1:GFP+ cells), that cx41.8 is enriched in caudal ECs, as well as in HSPCs and to a lesser extent in macrophages (Fig.5a-b). To investigate the loss of function of this connexin on HSC biology, we performed in situ hybridization for *runx1* and *cmyb* at different stages. We found that *cx41.8*^{-/-} mutants have normal emergence and specification of HSCs (*runx1* at 28hpf, *cmyb* at 36hpf). However, we found that *cx41.8*^{-/-} embryos present a significant decrease of HSCs in the CHT starting at 48hpf, before they are completely lost at 3.5dpf in the CHT. These results

confirm previous data in the mouse model, in which the deficiency of connexins induced a defect of HSCs expansion/maintenance in the bone marrow niche (*Singh AK, Cancelas JA. Int J Mol Sci. 2020*). Mechanistically, they showed that the connexins regulate stress-induced hematopoietic regeneration following the transfer of 5-FU-induced ROS from HSPCs to the bone marrow niche, and prevent ROS-p38-p16/INK4a mediated HSPC cell-death (*Singh AK, Cancelas JA. Int J Mol Sci. 2020*). To verify the hypothesis that the connexin-deficiency increases the level of ROS, we detected ROS (by Cell-ROX staining) in *cx41.8^{-/-}* comparing with wild type embryos at 60hpf. We found a significant increase of Cell-Rox positive cells in the CHT of *cx41.8^{-/-}* mutants (Suppl.Fig.12f-g). Next, we tried to rescue the loss of HSCs in the CHT of *cx41.8^{-/-}* mutants by GSH treatment. We treated *cx41.8^{-/-}* embryos for 12 hours (48-60hpf) with GSH 10 μ M and next measured *cmyb* at 60hpf in the CHT. We found that GSH treatment rescued the number of HSCs in *cx41.8^{-/-}* (Fig.5c-d). These results bring to light an important role of the connexin *cx41.8* during HSC expansion in the CHT.

These results confirm the data previously obtained by treating embryos with heptanol and carbenoxelone.

Several studies have previously demonstrated by Electron Microscopy cell-to-cell interactions between HSC and the surrounding niche cells (endothelial cells, stromal cells, and osteoblasts), which are essential for HSC localization, maintenance, and differentiation in mouse and zebrafish (Butler et al., *Cell Stem Cell* 2010; Tamplin O., et al., *Cell* 2015; Gori et al., *Stem Cell Transl. Med.* 2017). A growing body of work has detailed the importance of connexins Gap junction mediated intercellular communication (GJIC) in the regulation of signaling pathways required for HSC survival, proliferation, and fate decisions (Schajnovitz, et al., *Nature Immunol.* 2011). In particular, electron microscopy in the CHT has been performed to show direct interaction between HSPCs and endothelial cells, also known as “cuddling” (Tamplin, *Cell* 2015).

- There is no functional data for *gpx1a*, and the assumption that it is necessary at this stage of HSPC development is based on its expression pattern.

We removed the *gpx1a* data from this new version, and only mention this expression in the discussion as a data not shown.

- One of the “Key Points” states, “In the human fetal liver, IFI30 is expressed by macrophages and plays a similar role”, suggesting there is functional data from human fetal liver but there is not. Only expression data is presented.

We removed this key point, and added a key point about the *Cx41.8*.

- Pg.8, line 199 – “by in situ hybridization (Suppl. Fig.11a-b)” should specify that *cmyb* probe was used

We did.

REVIEWER COMMENTS

Reviewer #1 (Remarks to the Author):

The authors have worked to answer my concerns/doubts by providing new experiments/data. However, my concerns has not been resolved or argued convincingly and still remain. These revisions are not based on novel experiments or idea but rather on expected controls (new treatments, mutants, ISH). Some issues continue to exist (e.g. heart beating video of heptanol-specificity are poorly done; senesce vs cell death: it is not semantic problem but rather lack of acknowledge of basic biological concepts; ROS-related metabolites: what does it mean ?)

Overall, I don't feel comfortable in supporting the publication of this work.

Reviewer #3 (Remarks to the Author):

The authors have addressed previous comments and made appropriate revisions. However, there is extensive new data presented in this version that is beyond the scope of the original submission, including a new mutant model. This new data raises a number of new issues that are highlighted below and should be addressed. The primary concern is the revised model that includes cx41.8 but is still overstated and does not include sufficient supporting data.

Comments:

1. Background on cx41.8 is not sufficiently presented in the abstract or introduction, even though it is stated as a 'key point' of the paper.
2. Key point states, "Cx41.8 is necessary for communication between HSPCs and their niche by mediating exchange of ROS-related metabolites", however the term "communication" suggests signaling events that are not described in this paper. The statement should be revised and limited to oxidative stress.
3. The statement, "we examined the fate of HSPCs in connexin 41.8-/- mutant embryos" is misleading because there were no fate mapping studies.
4. Line 261 (Fig.14c-d) should be Supp. Fig.14c-d
5. This statement is misleading: "These results confirm the findings of a recent study, which reported that deficient connexins induce a defect of HSPCs expansion in the mouse bone marrow". The results from this study of zebrafish developmental hematopoiesis do not necessarily confirm results in adult mouse bone marrow. This should be revised.
6. What is the identity of the cell types that accumulate ROS in cx41.8-/- (Supp Fig 12F,G)? It was not confirmed that these are HPSCs.
7. The authors have not confirmed that the cx41.8 mutant phenocopies their chemical treatments. Cx41.8 mutant embryos could be treated with the heptanol drug to show that the phenotype is equivalent.
8. In Fig.5F the image is low magnification and it is not clear from the Z resolution if there is co-localization in the cmyb+ cells.
9. As in previous versions of this study, an assumption is being made that cmyb:gfp is HSPC-specific. The authors pointed out in the rebuttal that this was also assumed in other studies that they cited, however this has never been formally proven with transplant of cmyb:gfp+ cells. To address this question the authors could easily repeat the heptanol/ROS experiment using other transgenic lines (kdrl:gfp and mpeg:gfp) to exclude expression in these other cell types. The cmyb:gfp line could be crossed to mpeg:dsred to test for co-expression. In Supp. Fig. 14A,B there are ROS+/cmyb:gfp+ cells

and ROS-/cmyb:gfp+ cells, suggesting there is heterogeneity in this population. The use of the double transgenic *gata2b:KalTA4;UAS:ifi30* would not only express *ifi30* in HSPCs, but could express in hematopoietic progeny of expressing cells, such as macrophages (Supp Fig 14E,F).

10. In the model (Fig 6E, third panel, connexin mutant/inhibition), there is not sufficient data to support that the phenotype is HSPC-specific. The model will need to be revised if heptanol-induced Rox also accumulates in macrophages or ECs.

11. Fig. S2E-F The quality of the images is poor and the embryos are out of focus.

12. The supplementary movies are very out of focus and are not publication quality. How were Videos S3-8 quantified? How many replicates do they represent?

13. The authors should not equate activity and expression: "antioxidant activity of *ifi30* by qPCR and in situ hybridization (Suppl. Figure 5a-b-c), as its expression is augmented after oxidative stress"

14. The data presented in Fig 2C shows very small differences in relative gene expression level changes. Is it only 0.02 times higher in *kdrl+* ECs? However, the statement followed that, "*ifi30* was highly enriched in CHT-ECs (Fig.2c)."

Reviewer #1 (Remarks to the Author):

The authors have worked to answer my concerns/doubts by providing new experiments/data.

However, my concerns has not been resolved or argued convincingly and still remain. These revisions are not based on novel experiments or idea but rather on expected controls (new treatments, mutants, ISH).

Some issues continue to exist (e.g. heart beating video of hepatanol-specificity are poorly done; senesce vs cell death: it is not semantic problem but rather lack of acknowledge of basic biological concepts; ROS-related metabolites: what does it mean ?)

Overall, I don't feel comfortable in supporting the publication of this work.

We would like to thank reviewer1 for all the very constructive experiments that have helped to improve our manuscript. Indeed, thanks to your sugestions, we can now show that ifi30 behaves as an anti-oxidant molecule.

Reviewer #3 (Remarks to the Author):

The authors have addressed previous comments and made appropriate revisions. However, there is extensive new data presented in this version that is beyond the scope of the original submission, including a new mutant model. This new data raises a number of new issues that are highlighted below and should be addressed. The primary concern is the revised model that includes cx41.8 but is still overstated and does not include sufficient supporting data.

We have now addressed all your concerns, and we thank you for taking a lot of time to examine our manuscript, through all these versions. We really appreciated your input, and hope you will be satisfied with our answers.

Comments:

1. Background on cx41.8 is not sufficiently presented in the abstract or introduction, even though it is stated as a 'key point' of the paper.

Thank you for this advice. We now have included a small paragraph commenting on the role of Connexins (Cx43) in the mouse bone marrow, as indeed it was shown that

Cx-deficient HSPCs accumulated ROS, and underwent senescence (after multiple exposure to 5FU). We also now mention Cx41.8 in the abstract.

2. Key point states, “Cx41.8 is necessary for communication between HSPCs and their niche by mediating exchange of ROS-related metabolites”, however the term “communication” suggests signaling events that are not described in this paper. The statement should be revised and limited to oxidative stress.

We changed the sentence to “Cx41.8 is necessary for the exchange of ROS-related metabolites between HSPCs and their niche”. Indeed, our data do not suggest any communication between HSPCs and their niche, but rather just a transfer of ROS-related metabolites.

3. The statement, “we examined the fate of HSPCs in connexin 41.8^{-/-} mutant embryos” is misleading because there were no fate mapping studies.

We changed the sentence to “we examined the status of HSPCs in connexin41.8 (cx41.8) mutant embryos”.

4. Line 261 (Fig.14c-d) should be Supp. Fig.14c-d

Done.

5. This statement is misleading: “These results confirm the findings of a recent study, which reported that deficient connexins induce a defect of HSPCs expansion in the mouse bone marrow”. The results from this study of zebrafish developmental hematopoiesis do not necessarily confirm results in adult mouse bone marrow. This should be revised.

We agree with the reviewer and we have changed our statement in the corresponding results section. You can now read “The role of gap junctions has previously been described in the mouse bone marrow, where Cx43-deficient HSPCs showed high levels of exhaustion after 5FU-induced exposure¹⁸”

6. What is the identity of the cell types that accumulate ROS in cx41.8^{-/-} (Supp Fig 12F,G)? It was not confirmed that these are HPSCs.

To verify the identity of the ROS-accumulating cells in cx41.8^{-/-} embryos, we performed cell-ROX staining on cmyb:eGFP⁺ (wild type embryos) and cx41.8^{-/-};cmyb:eGFP embryos. We find that the majority of cell-ROX positive cells express GFP in cx41.8^{-/-} mutant embryos. Suppl.fig.13c (indicated with white arrows), phenocopying what we observed after heptanol treatment on cmyb:GFP embryos. Therefore, in the absence of a functional Cx41.8, ROS are sequestered in HSPCs, and cannot transit to the niche.

7. The authors have not confirmed that the cx41.8 mutant phenocopies their chemical treatments. Cx41.8 mutant embryos could be treated with the heptanol drug to show that the phenotype is equivalent.

To verify if the cx41.8 mutant phenocopies our chemical treatments (heptanol and carbenoxelone), we treated, as suggested by the reviewer, wild type and cx41.8^{-/-} embryos with heptanol (between 50-60hpf), and we performed in situ hybridization for cmyb at 60hpf. We scored the number of cmyb-positive HSPCs in the CHT. We find a significant decrease in cx41.8^{-/-} embryos, which is not more important after heptanol treatment (Suppl.Fig.15a-b-c). These results confirm that the cx41.8 mutant phenocopies treatment by a connexin inhibitor, and that cx41.8 therefore plays a major role in this mechanism.

8. In Fig.5F the image is low magnification and it is not clear from the Z resolution if there is co-localization in the cmyb⁺ cells.

We increased the magnification and added also the orthogonal projection of a cmyb⁺ cell that coexpresses cell-ROX. This is now represented in the figure 5e of this new version of the manuscript.

9. As in previous versions of this study, an assumption is being made that cmyb:gfp is HSPC-specific. The authors pointed out in the rebuttal that this was also assumed in other studies that they cited, however this has never been formally proven with transplant of cmyb:gfp⁺ cells. To address this question the authors could easily repeat the heptanol/ROS experiment using other transgenic lines (kdrl:gfp and mpeg:gfp) to exclude expression in these other cell types. The cmyb:gfp line could be crossed to mpeg:dsred to test for co-expression. In Supp. Fig. 14A,B there are ROS⁺/cmyb:gfp⁺ cells and ROS⁻/cmyb:gfp⁺ cells, suggesting there is heterogeneity in this population. The use of the double transgenic gata2b:KalTA4;UAS:ifi30 would not only express ifi30 in HSPCs, but could express in hematopoietic progeny of expressing cells, such as macrophages (Supp Fig 14E,F).

To verify if heptanol-induced Rox expression in macrophages or ECs, we performed heptanol/ROS experiment using other transgenic lines (mpeg:gfp and kdrl:gfp). First we treated mpeg1:GFP embryos with heptanol (50uM), and we compared with non-treated embryo. We find no Cell-ROX expression in mpeg1:GFP⁺ treated with heptanol. These results have now been added to the manuscript and are presented in the Suppl.Fig.15a. We also treated kdrl:GFP embryos with heptanol (50uM), and we compared with non-treated embryos. We find no Cell-ROX expression in kdrl:GFP⁺ treated with heptanol, and this has been added in Suppl.Fig.15b.

Next, to verify that cmyb:GFP⁺ cells are not macrophages, as suggested by this reviewer, we combined the cmyb:GFP and mpeg1:mcherry transgenic lines. We used

two different approaches, first by live-imaging we acquired images at 48hpf by confocal and we observed no co-expression (n=15). (See figure below).

cmyb:GFP mpeg1:mcherry

Next, we also analyzed by FACS this double transgenic line *cmyb:eGFP/mpeg1:mcherry* at 48hpf. Gated on live cells, after doublets exclusion. (See figures below).

The two transgenes do not overlap as scored by cytometry. The rare *mpeg1*+ cells that seem to express GFP (very low levels – orange gate) are likely due to residual to autofluorescence (quite common in macrophages). Previously, we have shown that HSPCs are present in the *cmyb:GFP* high fraction, after EHT in the aorta (Bertrand et

al., Nature, 2010). We have traced *cmyb*⁺ cells from the aorta and shown that they have lymphoid potential as they can home to the thymus (Bertrand et al., Development, 2008). We have characterized these cells by qPCR to show their hematopoietic progenitor signature. Of note, *cmyb*:GFP is also expressed by all cells in the neural tube, as well as by multi-ciliated cell on kidney tubules. Finally, we have also performed a few experiments experiments with *cd41*:GFP embryos (Suppl Fig 14c), which phenocopied the experiments performed with *cmyb*:GFP embryos. It was recently shown that *cd41*:GFP positive cells from 3 day-old embryos could repopulate adult *foxn1/casper* mutants, with a chimerism of 15% on average (Peng Lv et al, Stem Cell Reports, 2020; Figure 3).

Concerning the point that *gata2b* could be expressed in the hematopoietic progeny of expressing cells, such as macrophages, we performed qPCR for *gata2b* in different populations (fold change related to *kdrl*-head), and we find a high enrichment in *ikaros*⁺ cells, but *gata2b* is absolutely not expressed in *mpeg1*⁺ cells (See figure below). Moreover, a very recent study from the Wittamer lab, as shown that between 48 and 72 hpf, most of the macrophages present in the CHT do not derive from definitive hematopoiesis, but rather from primitive hematopoiesis (where *gata2b* is never expressed). They elegantly showed and quantified this by examining fate-mapping of the hemogenic endothelium (*kdrl*:CRE x *bactin*:Switch-DsRed) in the context of the *mpeg1*:GFP transgene (Ferrero, Development 2021, Fig 5-I, J, K).

10. In the model (Fig 6E, third panel, connexin mutant/inhibition), there is not sufficient data to support that the phenotype is HSPC-specific. The model will need to be revised if heptanol-induced Rox also accumulates in macrophages or ECs.

As we reported before, to verify if the cells that accumulate ROS in *cx41.8*^{-/-} were HSPCs, we performed cell-ROX staining on wild-type *cmyb*:GFP and *cx41.8*^{-/-}

;cmyb:GFP. We find that cell-ROX positive are cmyb:GFP⁺ cells (Suppl.fig.13c, indicated with white arrows).

We have performed heptanol/ROS experiment using other transgenic lines (mpeg:gfp and kdrl:gfp) to verify if heptanol induced ROS accumulation in macrophages or ECs. First we treated mpeg1:GFP embryos with heptanol (50 micromolar), and we compared with non treated embryo. We find no Cell-ROX expression in mpeg1:GFP⁺ treated with heptanol, in the new version of the manuscript is the Suppl.Fig.16a. Next we treated also kdrl:GFP embryos with heptanol (50 micromolar), and we compared with non treated embryo. We find no Cell-ROX expression in kdrl:GFP⁺ treated with heptanol, in the new version of the manuscript is the Suppl.Fig.16b.

These new results support the model that we proposed in fig.6e.

11. Fig. S2E-F The quality of the images is poor and the embryos are out of focus.

We changed the Suppl.fig.2e with new better images.

12. The supplementary movies are very out of focus and are not publication quality. How were Videos S3-8 quantified? How many replicates do they represent?

We added new supplementary videos with a better quality. Each video represents 9 replicates for each condition. This will also appear in the video legends.

For each condition to verify heart beating, we now have a brightfield view of the embryo, as well as a fluorescence microscopy video. When we treat with heptanol at 50uM (the concentration that impacts HSPCs), the videos are very similar to non treated embryos. Heart beatings are normal, and blood circulates normally. When we treat with 1mM, heptanol is toxic (embryos die after 30 minutes), and circulation is really slowed down, due to deficient heart beating.

13. The authors should not equate activity and expression: “antioxidant activity of ifi30 by qPCR and in situ hybridization (Suppl. Figure 5a-b-c), as its expression is augmented after oxidative stress”

We changed this sentence. “We found a significant increase in ifi30 expression following all pro-oxidant treatments tested (Suppl.Fig.5a-b). Ifi30 levels were particularly enhanced by the addition of GSSG, which was also confirmed by in situ hybridization(Suppl.Fig.5c)”.

14. The data presented in Fig 2C shows very small differences in relative gene expression level changes. Is it only 0.02 times higher in kdrl+ ECs? However, the statement followed that, “ifi30 was highly enriched in CHT-ECs (Fig.2c).”

In the new version of the manuscript, we present the data differently. The previous figure showed a delta-Ct analysis, where the expression of *ifi30* was compared to the expression of *ef1a* in each population tested. We now show a delta-delta Ct analysis where the *kdr1*:GFP+ head population serves as the reference population. The data is presented in Fig.2c, and shows that *ifi30* expression is highly enriched in *kdr1*:GFP+ from the tail region, as well as in *mpeg1*:GFP cell populations.

REVIEWERS' COMMENTS

Reviewer #3 (Remarks to the Author):

The authors have sufficiently addressed Reviewer #3 comments and suggestions, except the following that are minor revisions.

-Although many of the supplementary movies were improved, #1,2,9,10 are still very low resolution and out of focus.

-Fig S13C please add the white arrows to the green channel alone

-Typo in Line 267 "suggesting that Cx41.8 is tha major"

REVIEWERS' COMMENTS

Reviewer #3 (Remarks to the Author):

The authors have sufficiently addressed Reviewer #3 comments and suggestions, except the following that are minor revisions.

Thank you for acknowledging our work. We also have done our best to address the last minor points

-Although many of the supplementary movies were improved, #1,2,9,10 are still very low resolution and out of focus.

We deconvoluted our time-lapses differently for supplementary movies 1, 2 9 and 10. Instead of a 3D orthogonal view, where the tissue could seem out of focus, we used a maximum projection view. In particular, the 3D+t acquisitions were manually oriented to have the CHT horizontal. Then, when required, the presence of a 3D translational drift was corrected by detecting structures that are morphologically stable (therefore not intrinsically moving during the time course). Subsequently, a background subtraction was performed on the GFP channel. Finally, for display purposes, maximum intensity projections were used and the intensity of each channel of the images was set in order to saturate the bottom 1% and the top 0.1% of all pixel values. Movies were recorded using Lanczos' resampling with a kernel of size 3 and quantized over 8 bits per channel afterwards.

-Fig S13C please add the white arrows to the green channel alone

This has been added, and is now supplementary Figure 14 in this new version.

-Typo in Line 267 "suggesting that Cx41.8 is tha major"

Fixed, thanks.